# Single-cell analysis of hepatoblastoma identifies tumor signatures that predict chemotherapy susceptibility using patient-specific tumor spheroids

Hanbing Song [1,2,3,4,14], Simon Bucher [5,6,14], Katherine Rosenberg[6,7], Margaret Tsui[4,5,6], Deviana Burhan[5,6], Daniel Hoffman [6,7], Soo-Jin Cho [8,9], Arun Rangaswami[9,10], Marcus Breese[10], Stanley Leung [10], María V. Pons Ventura[10], E. Alejandro Sweet-Cordero[9,10], Franklin W. Huang [1,2,3,4,11,12,15] ✉, Amar Nijagal [4,6,7,9,13,15] ✉ & Bruce Wang [4,5,6,9,15] ✉

Pediatric hepatoblastoma is the most common primary liver cancer in infants and children. Studies of hepatoblastoma that focus exclusively on tumor cells demonstrate sparse somatic mutations and a common cell of origin, the hepatoblast, across patients. In contrast to the homogeneity these studies would suggest, hepatoblastoma tumors have a high degree of heterogeneity that can portend poor prognosis. In this study, we use single-cell transcriptomic techniques to analyze resected human pediatric hepatoblastoma specimens, and identify five hepatoblastoma tumor signatures that may account for the tumor heterogeneity observed in this disease. Notably, patient-derived hepatoblastoma spheroid cultures predict differential responses to treatment based on the transcriptomic signature of each tumor, suggesting a path forward for precision oncology for these tumors. In this work, we define hepatoblastoma tumor heterogeneity with single-cell resolution and demonstrate that patient-derived spheroids can be used to evaluate responses to chemotherapy.

[1]Division of Hematology and Oncology, Department of Medicine, University of California, San Francisco, San Francisco, CA 94143, USA. [2]Institute for Human Genetics, University of California, San Francisco, San Francisco, CA 94143, USA. [3]Bakar Computational Health Sciences Institute, University of California, San Francisco, San Francisco, CA 94143, USA. [4]Helen Diller Family Comprehensive Cancer Center, University of California, San Francisco, San Francisco, CA 94143, USA. [5]Division of Gastroenterology, Department of Medicine, University of California, San Francisco, San Francisco, CA 94143, USA. [6]The Liver Center, University of California, San Francisco, San Francisco, CA 94143, USA. [7]Division of Pediatric Surgery, Department of Surgery, University of California, San Francisco, San Francisco, CA 94143, USA. [8]Department of Pathology, University of California, San Francisco, San Francisco, CA 94143, USA. [9]The Pediatric Liver Center at UCSF Benioff Childrens' Hospitals, San Francisco, CA 94143, USA. [10]Division of Oncology, Department of Pediatrics, University of California, San Francisco, San Francisco, CA 94143, USA. [11]Department of Medicine, San Francisco Veterans Affairs Medical Center, San Francisco, CA 94121, USA. [12]Chan Zuckerberg Biohub, San Francisco, CA 94158, USA. [13]Eli and Edythe Broad Center of Regeneration Medicine, University of California, San Francisco, San Francisco, CA 94143, USA. [14]These authors contributed equally: Hanbing Song, Simon Bucher. [15]These authors jointly supervised this work: Franklin W. Huang, Amar Nijagal, Bruce Wang. ✉e-mail: franklin.huang@ucsf.edu; Amar.Nijagal@ucsf.edu; bruce.wang@ucsf.edu

Hepatoblastoma (HB) is the most common primary pediatric liver cancer, accounting for approximately 1% of all pediatric malignancies, and its incidence is rising[1]. Five-year survival for HB is among the lowest for childhood cancers, driven by the 20% of cases that are chemotherapy resistant or unresectable[2,3]. Current clinical risk stratification remains dependent on imaging and histological features at the time of diagnosis, with serum AFP as the only molecular marker[4]. There is an urgent need to improve the molecular characterization of HB to more accurately risk stratify patients. For patients with advanced HB, no effective treatment options exist outside of liver transplantation[5]. Progress in treating aggressive HB has been limited by the lack of models that reflect the heterogeneity of this tumor and that can be used to identify therapies[6]. The significant cellular heterogeneity observed in HB, both within and across patients[7], likely accounts for the limited utility of genomic studies from bulk tumor tissue for cancer staging[8–11]. Methods now exist for analyzing gene expression at the level of individual cells from dispersed neoplastic and normal tissues[12].

In this work, we use single cell RNA sequencing (scRNA-seq) to distinguish HB tumor cells from non-tumor cells, and to identify tumor cell signatures that may account for the heterogeneity observed in HB tumors. We also use HB patient-specific spheroids (PDS) to predict treatment response and identify therapeutic targets.

## Results

### Single-cell profiling of human pediatric HB reveals distinct clusters of tumor cells and tumor-associated populations

We established a workflow to isolate single cells from fresh pediatric HB tumor tissue at the time of surgical resection from nine patients (Fig. 1a). Our samples included both epithelial and mixed epithelial mesenchymal tumors, each of which exhibited a range of epithelial histology, though none had small cell undifferentiated features indicating high risk (Supplementary Data 1). Eight of the nine patients underwent chemotherapy prior to resection. Single cells from tumor and paired adjacent normal tissues were isolated for scRNA-seq analysis. One of the nine patients (Patient 5) had tumor extension to adjacent tissue, and therefore non-tumor tissue was unavailable. A total of 44,550 cells were captured, and 29,968 cells passed quality control (13,870 tumor, 16,098 non-tumor tissue) and were analyzed (Supplementary Fig. 1a–d and Supplementary Data 2). We used cut-off thresholds of >300 genes/cell and >500 transcripts/cell, and confirmed that clustering was unaffected when using a higher threshold (>500 genes/cell and 1000 transcripts/cell, Supplementary Fig. 2a–e). Using unbiased clustering and UMAP visualization, we identified 36 distinct clusters of cells (Supplementary Fig. 3a–c). We confirmed there was minimal patient to patient variability among cells from adjacent normal liver, indicating minimal inter-sample batch effect (Supplementary Fig. 4a, b).

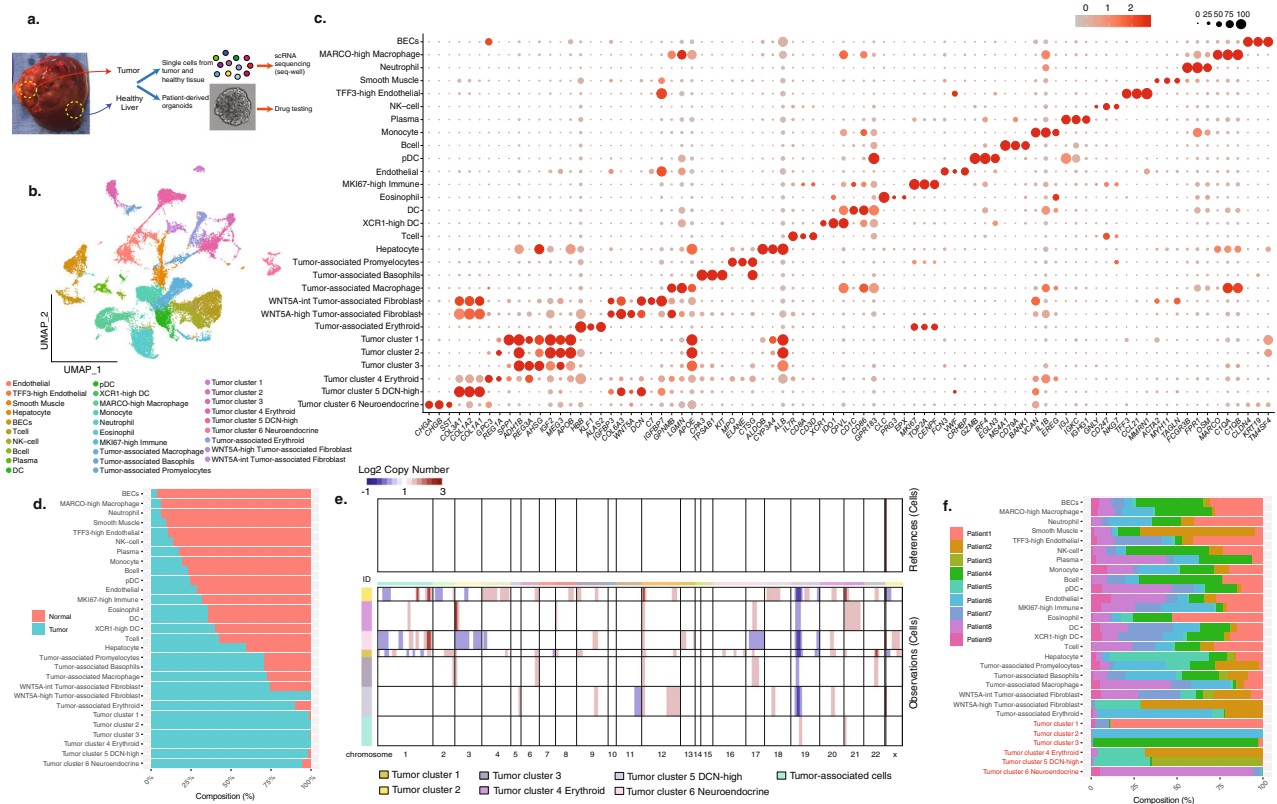

**Fig. 1 | Single-cell profiling of HB patients reveals distinctive tumor cell populations and tumor-associated populations. a** Flowchart of tissue processing of HB tumor and adjacent paired normal tissue samples for single-cell RNA sequencing and patient-derived spheroid culture. **b** Uniform manifold approximation and projection (UMAP) of 29,968 cells from nine HB patients, annotated by cell types. **c** Dot plot of all identified populations, each characterized by three known cell type markers. Average expression was indicated by the color gradient, and the percentage of marker expressed was represented by the dot size. **d** Stacked bar charts showing the contribution of the two sample types (normal and tumor) to each cell population, ranked by the contribution from normal samples. **e** Estimated copy-number alteration profile of all tumor and tumor-associated cell clusters using all the non-tumor and non-tumor-associated cells as reference. Chromosomes are labeled on the horizontal axis. Estimated copy numbers are shown in blue (deletion) and red (amplification) color bars. **f** Stacked bar charts show the contribution of the nine patients to each cell population.

To annotate these clusters, we performed differential gene expression analysis and identified 2130 differentially expressed genes (DEGs) (log2 fold change >1.0, adjusted p-value < 0.05) (Supplementary Fig. 3c and Supplementary Data 2). We manually selected three well-recognized and highly significant DEGs for each cluster and applied a descriptive nomenclature, defining 29 cell types (Fig. 1b, c). Seventeen of these clusters were composed of epithelial, stromal, or immune cell populations that are normally resident in the liver. Epithelial populations include hepatocytes and biliary epithelial cells. Stromal cell populations include two clusters of endothelial cells (*VWF +/MMRN1+)* differentiated by the level of *TFF3* expression, and one cluster of *MYH11+/ACTA2*+ smooth muscle cells. Immune cells identified include T-cells, NK-cells, monocytes, neutrophils, B-cells, plasma cells, eosinophils, macrophages, three clusters of dendritic cells (DC), and *MKI67* expressing proliferating immune cells (Fig. 1b, c). We used an automated cell type annotation tool SingleR (version 1.2.4) to validate our manual annotations (Supplementary Fig. 5)[13].

In the remaining 12 clusters, >60% of cells originated from tumor tissues suggesting these were either tumor clusters or tumor-associated clusters (Fig. 1d and Supplementary Fig. 4a). We designated the six "tumor clusters" based on their expression of high levels of known HB genes (Fig. 1c, Supplementary Fig. 6a, and Supplementary Data 2). Tumor clusters 1, 2, and 3 were characterized by high expression of the known HB tumor markers *REG3A, MEG3 and IGF2*[8]. Tumor cluster 4 expressed high levels of erythroid genes including *HBB*, *HBG1/2* and *ALAS2*. Tumor cluster 5 expressed high levels of fibroblast genes including *DCN* (Supplementary Fig. 6a–d). Tumor cell cluster 6 expressed neuroendocrine markers (e.g., *CHGA* and *CHGB*).

Despite >60% of cells within the remaining six clusters originating from tumor tissues, these clusters did not express HB markers and were designated as "tumor-associated clusters". They included three immune cell populations: a promyelocyte population, a tumor-associated macrophage (TAM) population expressing low levels of *MARCO*, and a basophil cluster. One tumor-associated cluster (*HBB +/ALAS2*+) expressed high levels of erythroid progenitor genes (*KLF1*) and was highly proliferative, which we annotated as tumor-associated erythroid cells[14]. We also identified two fibroblast populations differentiated by *WNT5A* expression level, which we annotated as *WNT5A*-intermediate and *WNT5A*-high tumor-associated fibroblasts (Supplementary Fig. 6b). While these two tumor-associated clusters had gene expression profiles similar to that of the *DCN*-high tumor cell cluster 5, they did not express HB genes including *GPC3, DLK1,* and *DKK1* (Supplementary Fig. 6a).

To validate the tumor cell annotations, we analyzed each tumor sample and adjacent normal liver for mutations in *CTNNB1* and found that seven of the nine patients had somatic mutations in *CTNNB1* (Supplementary Fig. 7a, corresponding Sanger sequencing traces are shown in Supplementary Fig. 8). We also analyzed copy number alterations (CNA) of each tumor specimen and found patient-specific differences in the five patients for whom CNA analysis was available (Supplementary Fig. 7b). These data indicate that the tumor specimens carried the somatic mutations and CNA that would be expected for HB. In order to specifically validate that the cells comprising the 6 tumor clusters were, in fact, composed of tumor cells, we performed inferCNV analysis to compare the estimated copy number profiles for all tumor and tumor-associated clusters in our dataset using the 17 non-tumor-associated clusters as reference[15]. Compared to reference cells, only the six tumor clusters exhibited significantly different CNV profiles, supporting their identity as tumor cells (Fig. 1e).

Interestingly, we found that four of the tumor cell clusters were unique to one patient while two were identified in two patients, indicating significant patient-specificity (Fig. 1f). By contrast, all 17 non-tumor-associated populations were present in every patient, with similar transcriptomes across patients (Fig. 1f and Supplementary Fig. 4b), further supporting the conclusion that the differences we identified between tumor clusters represent true transcriptomic differences between patients instead of batch effects.

## Five distinct tumor signatures may account for the heterogeneity observed in HB tumors

We performed sub-cluster analysis to further characterize HB tumor cells (*N* = 6244 cells), yielding eight tumor cell sub-clusters (Supplementary Fig. 9a–c and Supplementary Data 3). When investigating the composition of each sub-cluster, we found that patients 2 and 5 contributed to the erythroid tumor sub-cluster 6, consistent with the finding of extramedullary erythropoiesis in these two tumors (Supplementary Fig. 9b and Supplementary Fig. 10). Patients 3 and 5 contributed to the *DCN*-high tumor sub-cluster 7, consistent with fibrotic regions present in both tumors (Supplementary Fig. 9b and Supplementary Fig. 10). Closer examination of the DEGs showed that sub-clusters 1–5 shared expression of known HB markers including *GPC3, PEG10, REG3A, RELN*, and *PDK4* (Supplementary Fig. 9c)[8,9,16]. Notably, these sub-clusters were almost exclusively composed of cells from the four epithelial tumors (Supplementary Data 1 and Supplementary Fig. 10).

We investigated the similarity among tumor cells using a correlation heatmap with unsupervised clustering (Fig. 2a), demonstrating that the tumor cells fell into five distinct groups. The epithelial tumor cell populations 1–4 were similar to each other and formed a distinct group. A second group consisted primarily of cells from the only pure fetal HB specimen in our dataset. The three remaining groups consisted primarily of cells from the erythroid, *DCN*-high and neuroendocrine cell sub-clusters, respectively. Based on this, we generated five distinct HB signatures (Fig. 2a, b, Supplementary Fig. 9d, Supplementary Fig. 11a–e, and Supplementary Data 4). Each of these five groups contained cells from multiple tumor cell populations, and originated from multiple patients, suggesting that these five groups may represent common HB signatures shared across the tumors in our study (Supplementary Fig. 9d). Similarly, we calculated a signature score for each of the five HB signatures. With the exception of the Neuroendocrine signature that was almost exclusively expressed by patient 8, the remaining patients showed enrichment in at least two HB signatures (Fig. 2c–g, *p* < 2.2e−16, one-way ANOVA test). We further validated the five HB tumor signatures using fluorescence in situ hybridization (FISH) and immunofluorescence (IF) stains. We identified high levels of *IGF2* in patients enriched for Hepatoblastoma I and Hepatoblastoma II signatures (Fig. 2h, i, m), *GATA1* in patients enriched for the Erythroid-like signature (Fig. 2j, m), *POSTN* in patients enriched for the *DCN*-high signature (Fig. 2k, m), and *CHGA* in patient 8 who was enriched in the Neuroendocrine signature (Fig. 2l, m).

To further confirm the five HB tumor signatures identified in our analysis, we compared the five HB tumor signatures with published human liver bulk transcriptomic datasets[8,9]. We found that the Hepatoblast I signature was enriched in both fetal and adult liver genes while the Hepatoblast II signature was enriched only in adult liver genes (Supplementary Fig. 12a, *p* < 2.2e−16, two-sided Wilcoxon rank sum test). We then compared our results to published HB bulk RNA-seq datasets. First, we performed a pseudo-bulk RNA-seq analysis of our nine tumor samples and compared each tumor with HB bulk RNA transcriptomic datasets that have been recently reported[17,18]. The hepatic differentiation group reported by Hirsch et al.[17] and the hepatocyte group reported by Nagae et al.[18] were consistent with the four epithelial tumors in our analysis (patients 1, 4, 6, and 7). Despite lower mean expression of genes from the hepatocyte group, outlier cells did, however, have high expression of genes from this group among patients with non-epithelial subtypes of HB, supporting our findings that mean expression from bulk RNA sequencing does not capture the entire heterogeneity of a given tumor (Supplementary Fig. 12b). Similarly, we found that the tumors with mesenchymal components had the highest expression of genes from the

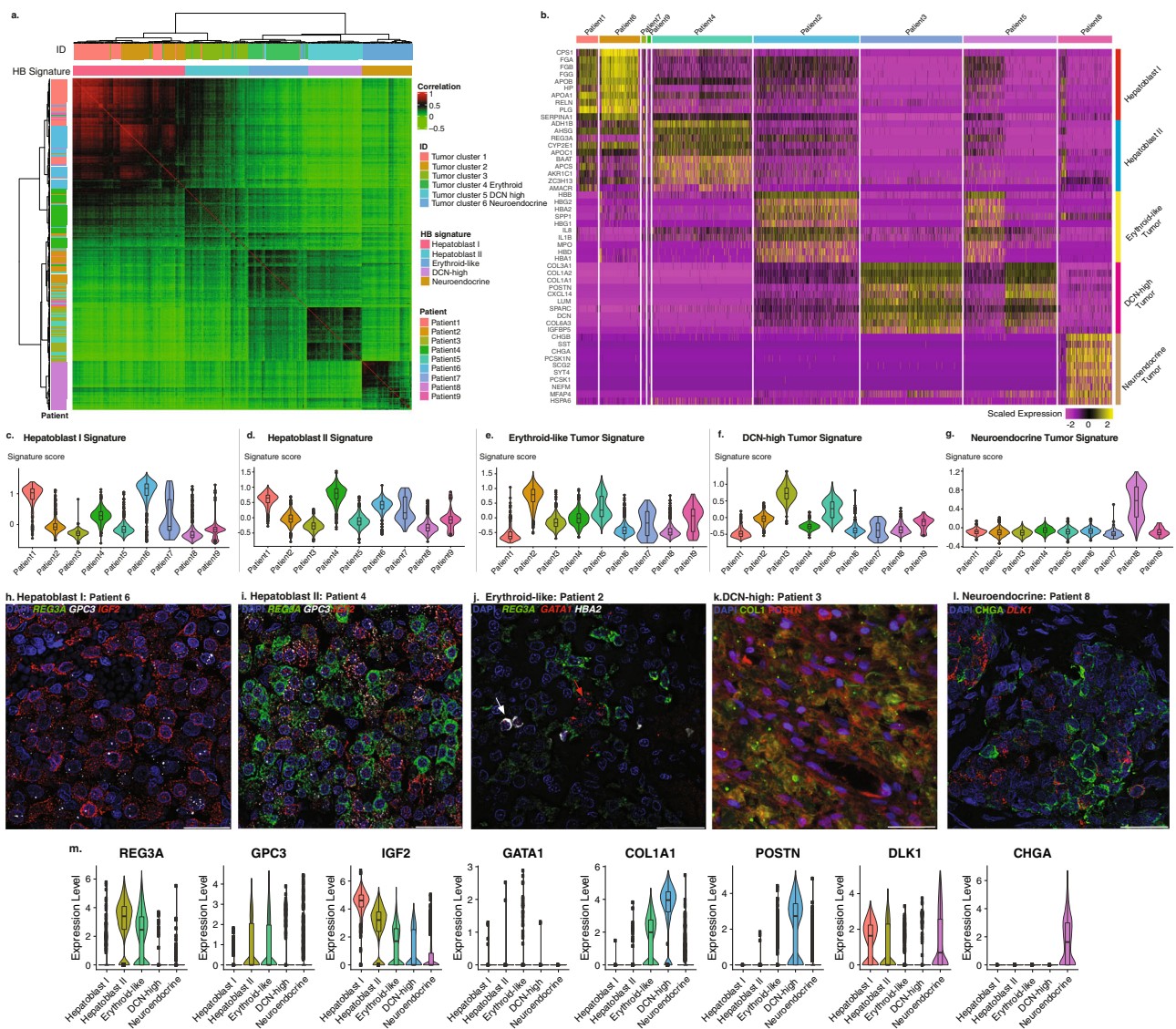

**Fig. 2 | Tumor cell analysis reveals five transcriptomically distinct tumor signatures detected within the nine HB patients. a** Correlation heatmap of the tumor cells by tumor cell clusters, tumor signatures (column annotations), and patients (row annotation). Correlation was shown by the color gradient. Hierarchal clusters were illustrated by dendrograms. **b** Heatmap of top 10 most differentially expressed genes of each HB signature. Scaled expressed levels are shown by the color bar. **c** Violin and box plot of the computed c. Hepatoblast I, **d** Hepatoblast II, **e** Erythroid-like, **f** DCN-high, and **g** Neuroendocrine tumor signature scores for all nine patients ($N = 6244$ cells). **h** FISH staining for *REG3A* (green), *GPC3* (white), and *IGF2* (red) of patient 6 tumor tissue (Hepatoblast I), and **i** patient 4 tumor tissue (Hepatoblast II). **j** FISH staining for *REG3A* (green), *HBA2* (white), and *GATA1*(red), of patient 2 tumor tissue showing Erythroid-like cells (red arrow) and tumor-associated erythroid cells (white arrow). **k** Immunofluorescence staining for *COL1* (green) and *POSTN* (red) of patient 3 tumor tissue (*DCN*-high). **l** Immunofluorescence staining for *CHGA* (green) and FISH staining for *DLK1* (red) of patient 8 tumor tissue (Neuroendocrine). **m** Violin plots of individual marker genes presented on panels (**h**–**l**). Scale bar = 30 μm. The box plots present the 25th percentile, the median, the 75th percentile, and outlying or extreme values. The whiskers of the box plots extend to a maximum of 1.5 times the size of the interquartile range. $N = 6244$ cells for all box plots. Source data are provided as a Source Data file.

mesenchymal group reported by Hirsch et al.[17] and Nagae et al.[18] (Supplementary Fig. 12b). Other published categories like the progenitor group were only minimally enriched in one patient in our dataset. Next, we compared the five HB signatures identified in this study to the same previously published HB groups. We found that the Hepatoblast I and Hepatoblast II signatures had strong associations with the hepatocyte and hepatic differentiation groups. We also observed associations between the previously published mesenchymal group, and the Erythroid-like and *DCN*-high signatures, and between the proliferative/proliferation group and the Neuroendocrine signature (Supplementary Fig. 12c). Taken together, our data are consistent with previously published HB tumor groups obtained from bulk RNA-seq datasets; however, the five HB signatures identified in

our analysis incorporates additional tumor heterogeneity that is not captured in the existing groups defined by bulk RNA datasets.

We also performed gene ontology analysis of the five HB signatures and found that the Hepatoblast I and II signatures were enriched for metabolic processes typically found in hepatocytes, the Erythroid signature was enriched for immune and detoxification processes, the *DCN*-high signature was enriched for bone and extracellular matrix processes, and the Neuroendocrine signature was enriched for neuronal processes (Supplementary Fig. 12d). This analysis further supports our identification and characterization of distinct HB signatures.

We questioned whether the five HB tumor signatures could help predict tumor aggressiveness. We first used previously published

predictive transcriptomic data and performed pseudo-bulk RNA-seq analysis of our nine tumor samples, and found that none of our nine pseudo-bulk tumor transcriptomes were significantly enriched for either the low-risk rC1 or the high-risk rC2 gene sets (Supplementary Fig. 12e)[8]. Next, we repeated the analysis but used the five HB tumor signatures and found that Hepatoblast I and II showed significant enrichment for the low-risk rC1 gene set compared to other signatures ($p < 2.2\text{e}{-}16$, two-sided Wilcoxon rank sum test), and the Neuroendocrine tumor subtype was enriched in the high-risk rC2 gene set ($p = 5.37\text{e}{-}3$, two-sided Wilcoxon rank sum test, Supplementary Fig. 12f). Interestingly, patient 8, who contributed all HB tumor cells harboring the Neuroendocrine signature, developed early relapse of HB within 12 months of surgical resection (Supplementary Data 1). These data suggest that the five HB tumor signatures may inform risk stratification for HB tumors.

Finally, we asked if the five HB tumor signatures could account for the heterogeneity within each tumor. First, we characterized the degree of heterogeneity by sub-clustering tumor cells for each patient (Supplementary Fig. 13a) and found that HBs exhibit a range of heterogeneity within each tumor consistent with the observed pathology. Patient 4 tumor cells were the most homogeneous, consistent with its pure fetal histology, whereas patient 8 tumor cells demonstrated the greatest degree of heterogeneity, consistent with its mixed epithelial-mesenchymal histology. We used FISH to validate the predominant sub-clusters within each tumor (Supplementary Fig. 13a). We also generated a heatmap and showed that the five signatures corresponded well to most of the patient-specific tumor clusters (Supplementary Fig. 13b). This suggests that intratumor heterogeneity may be accounted for by the five HB tumor signatures. We also detected subclusters that did not show an enrichment in any of the five tumor cell signatures, raising the possibility of other signatures in HB tumors (Supplementary Fig. 13b).

### Low *MARCO* expression distinguishes HB tumor-associated macrophages from macrophages in normal tissue

We assessed the composition of the tumor microenvironment in HB and questioned whether differences in the tumor microenvironment were associated with the heterogeneity observed among tumor cells. We examined the immune and stromal cells in our dataset that were enriched within HB tumors (Fig. 3a, b). Three immune cell populations were enriched within tumor, including macrophages, pro-myelocytes, and basophils (Fig. 3c). Subset analysis of each of these three populations showed that tumor-associated macrophages (TAM) were the only cell type that exhibited significant transcriptomic differences compared to non-tumor macrophages (Fig. 3d and Supplementary Fig. 14a–d). We compared the DEGs between macrophages from HB tumor tissue and adjacent non-tumor liver, and found that attenuated expression of the scavenger receptor, *MARCO*, distinguished macrophages found in tumors (*MARCO*^Low^) from those isolated from adjacent normal liver (*MARCO*^Hi^) (Fig. 3e, f). Notably, all nine patients contributed to the *MARCO*^Low^ cluster in HB tumors (Supplementary Fig. 14e). We identified several DEGs that were upregulated in tumor-associated *MARCO*^Low^ macrophages including the pro-tumorigenic chemokine *CCL18*[19] and genes known to promote cell proliferation, invasion, and migration (glycoprotein nonmetastatic melanoma protein B, *GPNMB*[20]) (Fig. 3g). Gene ontology pathway analysis demonstrated that *MARCO*^Low^ TAMs had higher expression of genes involved with antigen presentation, chemotaxis, and processes related to protein production (Fig. 3h). We validated the low expression of *MARCO* on HB TAMs using FISH, demonstrating the absence of *MARCO* on *CD163*^+^ macrophages in HB tumor tissues (Fig. 3i). Pseudotime analysis of the two macrophage populations and the monocyte population demonstrated that the *MARCO*^Low^ TAMs were transcriptomically more similar to monocytes than *MARCO*^Hi^ liver-

resident macrophages (Fig. 3j). These data support the origin of *MARCO*^Low^ TAMs as monocyte-derived macrophages.

We next asked whether *MARCO*^Low^ TAMs interact differently with the five HB tumor signatures. We identified candidate ligand-receptor interactions between *MARCO*^Low^ TAMs and four of the five HB tumor signatures. These distinct *MARCO*^low^ TAM interactions include PDGF signaling with cells expressing the *DCN*-high signature, the adhesion molecule CD44 with cells expressing the Erythroid-like signature, members of the TNF receptor superfamily with cells expressing the Hepatoblast I tumor signature, and plexin proteins (PLXNB2) with cells expressing the Neuroendocrine tumor signature (Supplementary Fig. 14f). Collectively, these findings indicate that *MARCO*^Low^ TAMs are a transcriptomically distinct population in HB tumors that express several pro-tumorigenic genes. These data also suggest that *MARCO*^Low^ TAMs may have unique interactions with HB tumor cells and may guide the different tumor signatures observed in HB tumors (Supplementary Fig. 14f), though further work is needed to validate the interactions between *MARCO*^Low^ TAMs and HB tumor cells.

### Developmentally restricted erythroid progenitor cells associated with hepatoblastoma

The identification of an HB-associated erythroid population suggests that in postnatal livers, the tumor microenvironment of HB may maintain a fetal liver-like niche that results in persistence of erythroid progenitor cells. In our dataset, the HB-associated erythroid population was seen in three patients (patients 2, 5, and 6). We first asked if HB-associated erythroid cells from these three patients had similar transcriptomes as human fetal liver erythroid cells. We integrated HB-associated erythroid cells with fetal liver erythroid cells from a reference dataset[14]. HB-associated erythroid cells shared gene expression profiles with human fetal liver erythroid cells and expressed markers from all three developmental stages, indicating that early, mid and late erythroblasts were present in the HB-associated population (Fig. 4a, b).

Next, we projected markers of fetal liver erythroid developmental stages with fetal erythroid markers *HBA2*, *HBB*, *HBG1*, and *HBG2* on HB-associated erythroid cells from patients 2, 5, and 6. We discovered that only cells from patients 2 and 5 expressed early erythroid markers whereas cells from patient 6 expressed mid and late erythroid genes (Fig. 4c). We confirmed this by analyzing HB-associated erythroid cells separately and showed that cells from patients 2 and 5 were predominantly early stage erythroids, whereas those from patient 6 were primarily mid to late stage erythroid cells (Supplementary Fig. 15a).

We examined whether erythropoiesis in HB progresses normally. Pseudotime analysis with the reference fetal erythroid cells (Supplementary Fig. 15b) and tumor-associated erythroid cells showed that cells from patients 2 and 5 were arrested at the early erythroblast stage[21]. Cells from patient 6, on the other hand, were at mid and late erythroblast stages (Supplementary Fig. 15c).

Erythropoiesis in the fetal liver occurs within specialized niches composed in part by erythroblastic island macrophages and depends on intercellular interactions between the niche and erythroid progenitor cells that differ based on the specific stage of erythropoiesis[14,22]. We hypothesized that TAMs and tumor cells from patients 2, 5, and 6 form a niche and maintain tumor-associated erythroid cells at their respective developmental stage. Significant ligand-receptor interactions ($p < 0.05$) identified via CellPhoneDB (Version2.1.4) between these potential niche cell populations, and the corresponding HB-associated erythroid cells demonstrated that TAMs and tumor-associated cells from patients 2 and 5 expressed early erythroid niche signals, while those from patient 6 expressed late erythroid niche signals (Fig. 4d)[23]. We confirmed one of the TAM and tumor-associated erythroid cell interactions from our ligand-receptor analysis by using FISH to localize *VCAM1*-expressing *CD163*^+^ TAMs that are adjacent to *ITGA4*-expressing tumor-associated erythroid cells

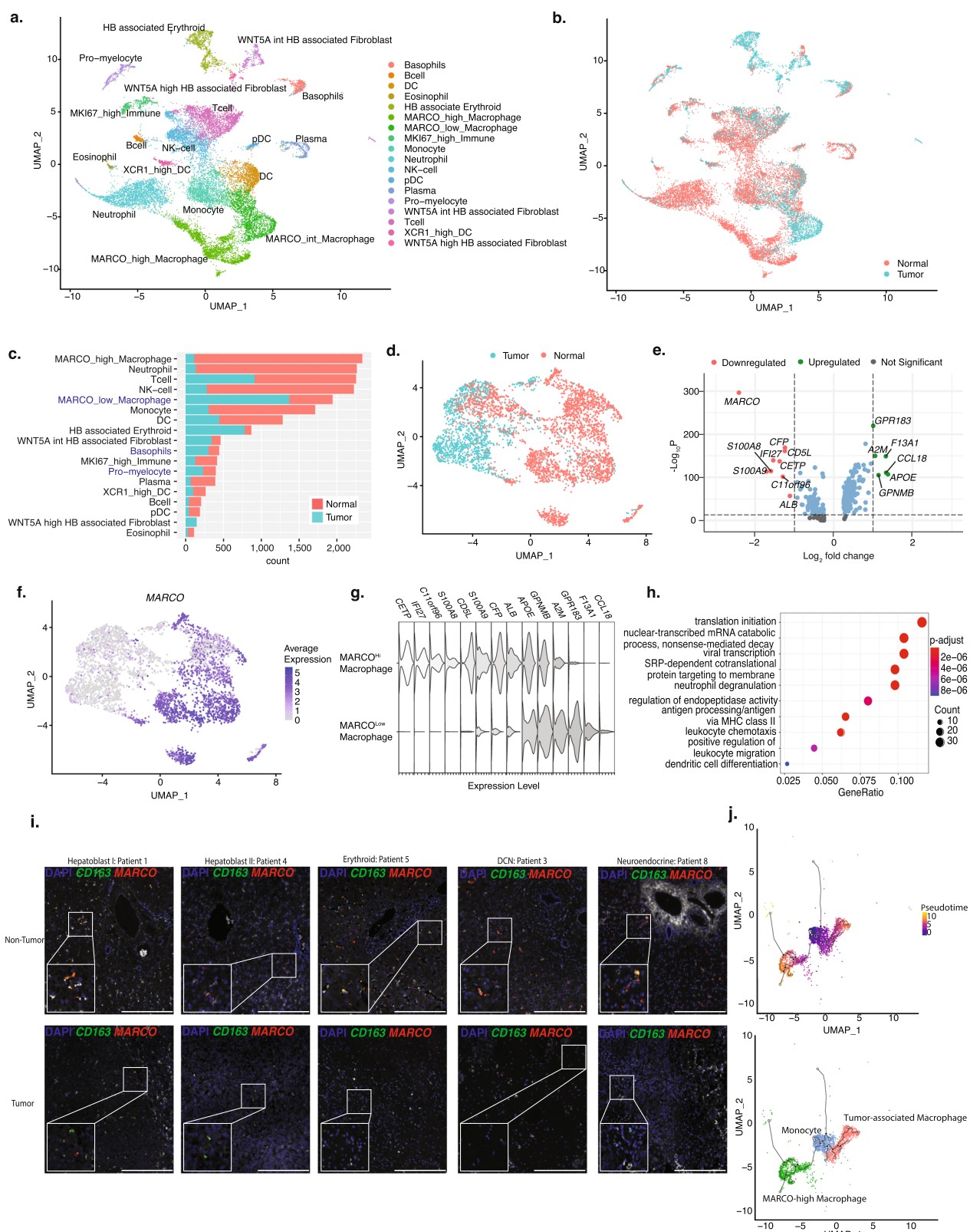

(Fig. 4e)[24]. We also examined interactions between tumor cells and tumor-associated erythroid cells. We identified and validated adjacent *IGF2*-expressing tumor cells with *IGF1R*-expressing tumor-associated erythroid cells in patient 6 (Fig. 4g)[25]. We also identified *SPP1* expressing tumor cells with *ITGA4*-expressing tumor-associated erythroid cells in patient 2 (Fig. 4h)[26]. These data demonstrate that both TAMs and tumor cells interact with tumor-associated erythroid cells to maintain fetal erythropoiesis at different developmental stages, supporting the hypothesis that HB tumor heterogeneity is due in part to the fetal stage at which the tumor arises[8,11,27].

Next, we examined whether tumor cells retained transcriptomic signatures corresponding to specific fetal liver developmental time points. We integrated tumor cells with a reference fetal liver hepatoblast dataset but found little overlap, indicating that tumor cells had

**Fig. 3 | Low MARCO expression distinguishes HB TAMs from macrophages in normal tissue.** UMAP of all immune cells and tumor-associated stromal cells from all 9 patients annotated by **a** cell type or **b** sample type. **c** The quantities of each cell type in both tumor and adjacent normal tissue. **d** UMAP of all macrophages from all 9 patients annotated by sample type. **e** Volcano plot showing top differentially expressed genes between macrophages found in tumors compared to those found in adjacent normal liver tissue. **f** Expression of the scavenger receptor, MARCO, distinguishes TAMs from normal liver macrophages. **g** Violin plots showing expression of the genes identified in the above volcano plot in MAR-COLow TAMs and normal liver MARCOHi macrophages. **h** Gene ontology terms that are associated with the genes differentially expressed in MARCOLow macrophages. Statistical significance levels (over-expression statistical test) are indicated in the color bar. **i** FISH staining for CD163 (green) and MARCO (red) of non-tumor and tumor tissue for patient 1 (Hepatoblast I), patient 4 (Hepatoblast II), patient 5 (Erythroid), patient 3 (DCN), and patient 8 (Neuroendocrine) showing near-total absence of MARCO in tumor tissue. **j** Computed pseudotime trajectory and anno-tated UMAP of monocytes, tumor-associated macrophages, and MACROHi macrophages.

gene expression patterns distinct to hepatoblasts (Supplementary Fig. 16a–c)[16,28].

We noticed that patients 2 and 5 had tumor cells with expression of erythroid genes but patient 6 did not. We ensured that the presence of erythroid genes in these cells was not due to ambient RNA contamination or doublets by using DecontX (Version 1.4.7)[29] and DoubletFinder (Version 2.0.3)[30] in our analytical pipeline. Tumor cells expressing the Erythroid-like signature in patients 2 and 5 expressed early-stage erythroblast markers, but had lower expression of mid and late stage erythroid markers *HBG1, HBG2, HBA2*, and *HBB* than tumor-associated erythroid cells. Tumor cells from Patient 6 showed little to no expression of most erythroid markers, consistent with our previous analysis that Patient 6 tumor cells were enriched in the Hepatoblast I signature (Supplementary Fig. 16a, b). Finally, we examined the significant ligand-receptor interactions between TAMs and tumor cells and found erythropoiesis signals were only present between TAMs and tumor cells for patients 2 and 5 (Supplementary Fig. 16b), suggesting that erythropoietic signaling may be involved in the oncogenesis for at least a subset of HBs.

## Patient-derived-spheroids maintain patient-specific features from freshly isolated parent cells

We generated five patient-derived spheroids (PDS) from fresh HB tumor samples. The success rate was greater when PDS were grown from freshly isolated cells than from cryopreserved cells (Supplementary Data 5). Four of the five PDS were maintained for more than 15 passages (>6 months, Supplementary Fig. 18a). PDS had similar proliferation rates that remained stable over time (Supplementary Fig. 18b). The PDS from patient 9 had two distinct cell populations corresponding to non-tumor BECs and fibroblasts, neither of which could be maintained long-term in our culture conditions (Supplementary Fig. 18c). PDS from different patients had distinct morphologies, which persisted with long-term culture (Fig. 5a and Supplementary Fig. 18a). Interestingly, patient 2 PDS formed large lumens, reminiscent to cholangioblastic features observed in patient 2 tumor tissue. Similarly, the dense nature of PDS from patient 8 is similar to the dense features in patient 8 tumor tissue (Supplementary Fig. 18d). Notably, gene expression patterns identified in each patient's tumor were also observed using FISH and IF of corresponding spheroids (Supplementary Fig. 18d).

To further characterize the PDS, we performed scRNA-seq on spheroids at both early, and late passages. PDS have broad gene expression differences compared to freshly isolated tumor cells, but retained their transcriptomic differences relative to other PDS even with long-term culture, indicating that they maintained patient-specific features (Supplementary Fig. 19a and Supplementary Data 6). All PDS had high expression of HB genes and retained their parent tumor mutations in *CTNNB1* (Fig. 5b, c), indicating that they grew from tumor cells. To directly assess which specific cell population each PDS grew from, we generated a signature score for each cell population in the tumor tissue and found that PDS were most similar to the tumor cell population in each case, even after long-term culture (Fig. 5d and Supplementary Fig. 19b–f). We could not identify a clear cell of origin for PDS from patient 2, likely because we did not capture the full spectrum of tumor cells in this patient (Supplementary Fig. 19d).

PDS from patient 7 had two distinct cell populations, a large population of spheroid cells similar to tumor cells, and a small population similar to fibroblasts. This is consistent with the finding of fibroblast-appearing cells in early passages for patient 7, which disappeared with long-term culture (Supplementary Fig. 19d).

We next asked if PDS retained gene expression of any of the five HB tumor cell types we identified. Since we did not capture the parent cells for PDS from Patient 2 and very few tumor cells were captured from patient 7, we focused our analysis on patients 6 and 8. We analyzed the tumor cells from patients 6 and 8 and generated a signature score for each subpopulation (Supplementary Fig. 19c and Supplementary Fig. 20a, b). PDS from patient 6 is most similar to the population of tumor cells that expressed the Hepatoblast I signature. patient 8 PDS is most similar to the population of tumor cells with the Neuroendocrine signature (Supplementary Fig. 19e and Supplementary Fig. 20c, d). This suggests that each PDS grows from a specific subpopulation of tumor cells, which is consistent with the relatively homogeneous transcriptome profile of PDS. We measured the gene signature score for each PDS relative to the five HB tumor signatures and showed that PDS from patients 6, 7, and 8 likely originated from cells expressing one of the five HB tumor signatures we identified, but patient 2 did not (Fig. 5e).

## Pharmacologic testing of patient-derived HB spheroids reveals HB tumor cell type specific treatment responses

We examined whether PDS can be used to test drug-sensitivity in a patient-specific manner. We treated PDS with five chemotherapeutics commonly used for HB and found that spheroids from patient 7 were the most sensitive to the drugs tested, while PDS from patients 2 were the most resistant (Fig. 6a–c and Supplementary Fig. 21a). Drug sensitivity is partly dependent on the cell's ability to metabolize and efflux the agent. We measured expression levels of genes known to be important for efflux and metabolism of platinum-based drugs, etoposide, vincristine, and 5-FU (https://www.pharmgkb.org/pathway/PA150653776, https://www.pharmgkb.org/pathway/PA150981002, https://www.pharmgkb.org/pathway/PA2025, https://www.pharmgkb.org/pathway/PA150981002) and found that in both tumor cells and spheroids, patient 2 had the highest expression of these genes while patient 7 had the lowest (Fig. 6b, c and Supplementary Fig. 22a–f).

Next, we asked whether PDS can be used to identify treatment targets for HB. We looked for known molecular pathways that are important for HB pathogenesis and that are differentially expressed within HB tumors and corresponding PDS. We found that the YAP signaling pathway was highly expressed in patients 2 and 7 cells (Fig. 6d). We treated PDS with the YAP1 inhibitor verteporfin and found that patient 7 PDS were most responsive to YAP1 inhibition whereas patient 2 PDS were not (Fig. 6d and Supplementary Fig. 21b). To explain this discordance, we looked at the expression of genes known to be important for efflux and metabolism of verteporfin and found that patient 2 had the highest expression of these genes (Fig. 6b and Supplementary Fig. 22d)[31]. Taken together, these results demonstrate that drug sensitivity profiles of PDS are globally linked to the

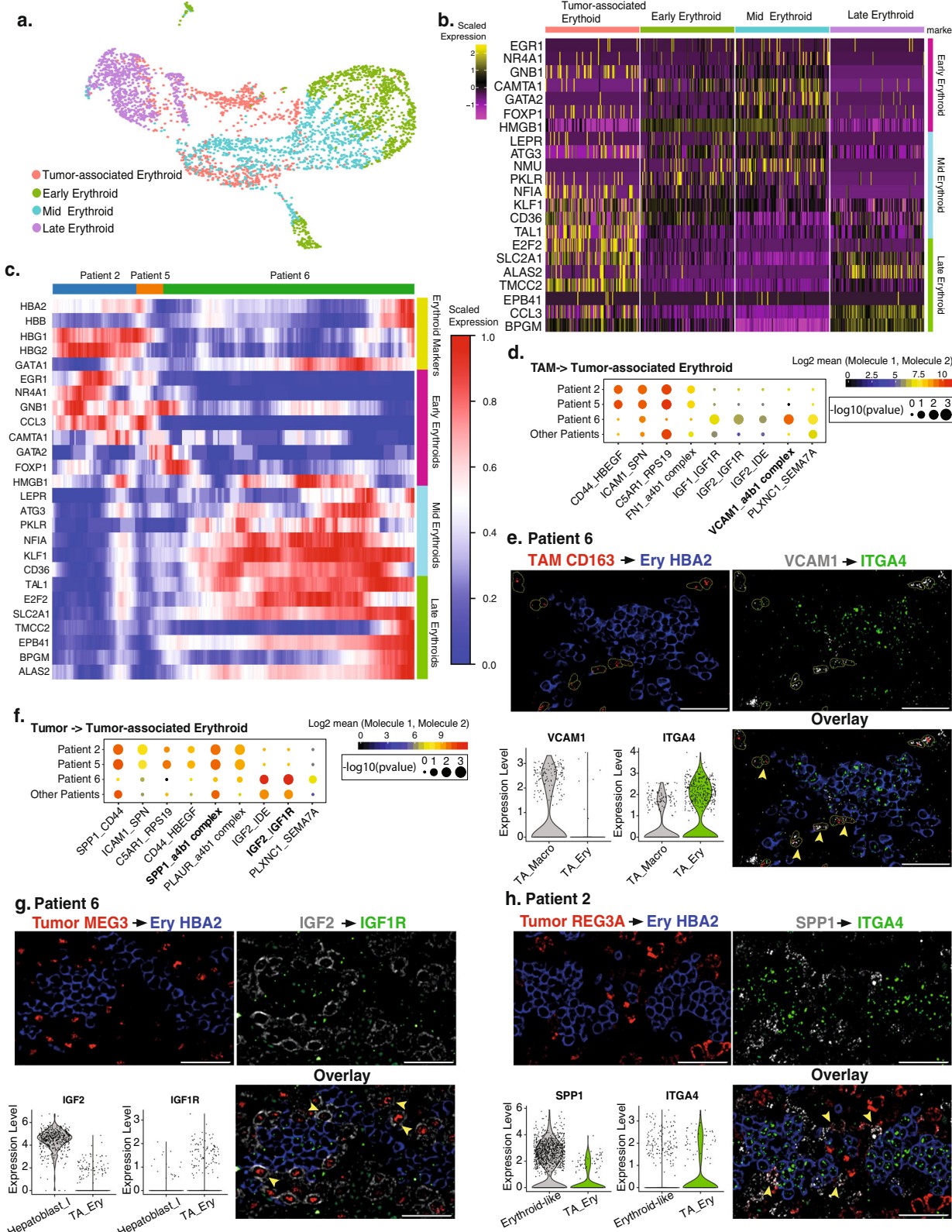

expression of drug efflux and metabolism genes in the tumor cell and PDS (Fig. 6b, c).

Since patient 8 had an early relapse, we looked specifically at signaling pathways differentially expressed in patient 8 tumor and PDS. We found patient 8 PDS expressed genes in the LIN28B pathway and exhibited sensitivity to the LIN28B pathway inhibitor, LIN28B-1632 (Fig. 6e and Supplementary Fig. 21c). Finally, we also observed

that PDS from patients 6 and 8 (both having relapsed) had a lower level of expression of proteasome coding genes (Fig. 6f), thereby predicting heightened sensitivity to the proteasome inhibitor, bortezomib[32,33]. Treatment of PDS from patients 6 and 8 demonstrated increased sensitivity to bortezomib compared to the other PDS (Fig. 6f and Supplementary Fig. 21d). These data indicate that the transcriptomic profile of HB PDS can be used to predict drug

**Fig. 4 | HB maintains erythropoiesis at three distinct developmental stages.**
**a** Integrated UMAP of fetal liver erythroid cells and tumor-associated erythroid cells. **b** Heatmap of the three developmental stages of fetal-liver erythroid marker expression for tumor-associated erythroid cells and reference fetal liver erythroid population. Scaled expressed levels are shown by the color bar. **c** Supervised heatmap showing expression of canonical erythroid markers and representative markers of three erythroid developmental stages. **d** Selected ligand-receptor interactions from TAMs to tumor-associated erythroid cells for patients 2, 5, 6, and other patients. Mean expressions of ligand and receptor pairs are shown in the color bar. Statistical significance levels (random permutation test) are indicated by the marker size. **e** Stainings for TAMs to tumor-associated erythroid cells ligand-receptor interactions by FISH with *CD163* (red), *HBA2* (blue), *VCAM1* (white), and *ITGA4* (green) on patient 6 tumor tissue, and violin plot of *VCAM1* and *ITGA4* for TAMs and tumor-associated erythroids from patient 6 tumor tissue. Arrows show TAMs expressing *VCAM1* next to tumor-associated erythroids expressing *ITGA4*.

**f** Selected ligand-receptor interactions between tumor cells and tumor-associated erythroid cells for patients 2, 5, 6, and other patients. Mean expression of ligand and receptor pairs are shown in the color bar. Statistical significance levels (random permutation test) are indicated by the marker size. **g** Stainings for tumor cells to tumor-associated erythroid cells ligand−receptor interactions by FISH with *MEG3* (red), *HBA2* (blue), *IGF2* (white), and *IGFR1* (green) on patient 6 tumor tissue, and violin plot of *IGF2* and *IGFR1* for tumor Hepatoblast I and tumor-associated erythroids from patient 6 tumor tissue. Arrows show tumor cells expressing *IGF2* next to tumor-associated erythroids expressing *IGFR1*. **h** Stainings for tumor cells to tumor-associated erythroid cells ligand−receptor interactions by FISH with *REG3A* (red), *HBA2* (blue), *SPP1* (white), and *ITGA4* (green) of patient 2 tumor tissue, and violin plot of *SPP1* and *ITGA4* for tumor Erythroid-like and tumor-associated erythroids from patient 2 tumor tissue. Arrows show tumor cells expressing *SPP1* next to tumor-associated erythroids expressing *ITGA4*. Scale bar = 35 μm.

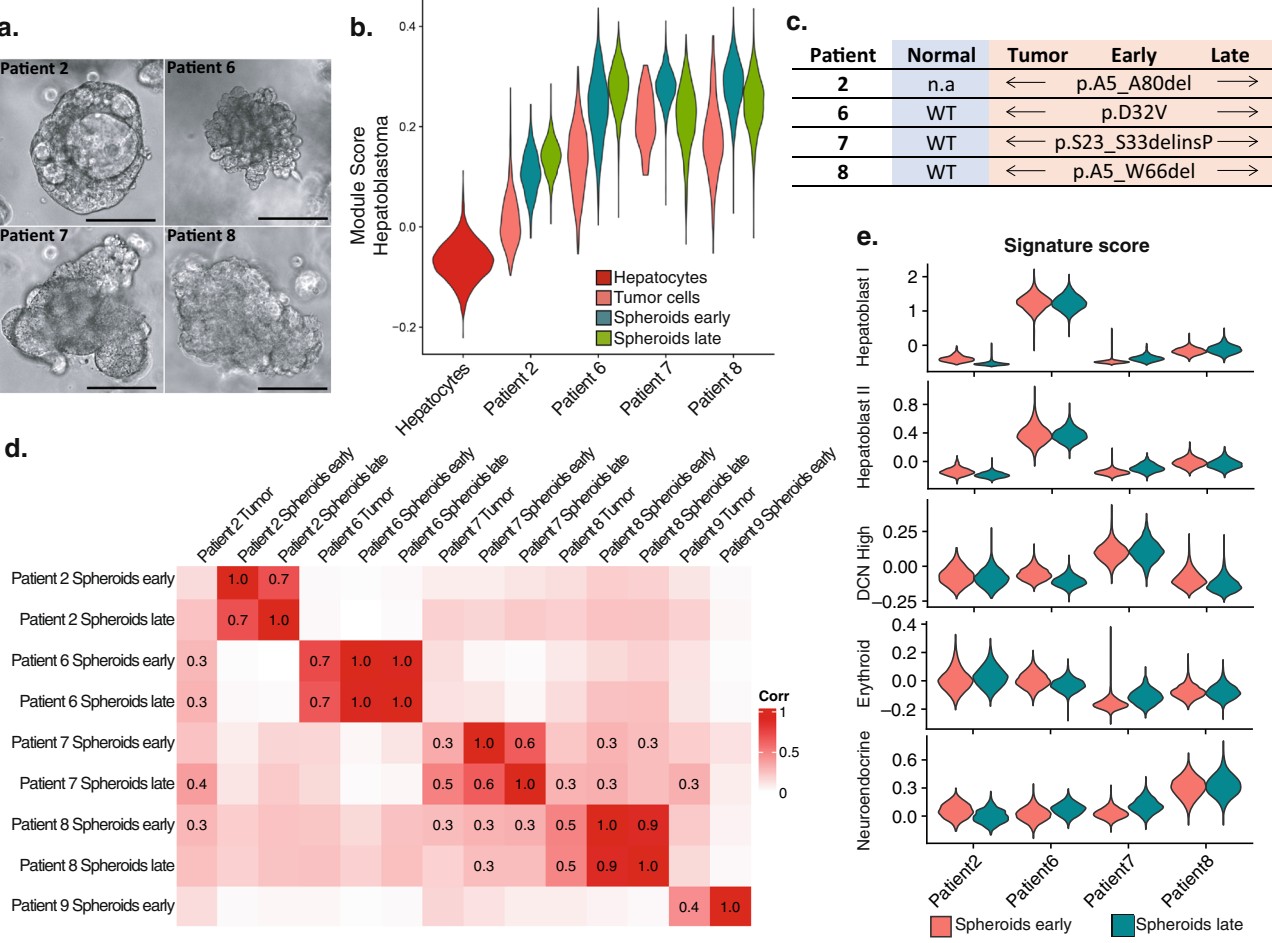

**Fig. 5 | Patient-derived-spheroids maintain patient-specific features from freshly isolated parent cells.** **a** Brightfield images of tumor spheroids from patients 2, 6, 7, and 8 at early passage. Scale bar = 100 μm. **b** Violin plots of KEGG Cairo_Hepatoblastoma_UP signature score of tumor cells (patient 2) or spheroid parent cells (patients 6, 7, and 8) or spheroids from patients 2, 6, 7, and 8 at an early passage (passage 2 for patients 2, 8, and passage 3 for patients 6 and 7) and at a late passage (passage 10 for patients 2, 6, and 8, and passage 11 for patient 7). Signature scores for all hepatocytes from all patients are represented as a reference. **c** Exon 3 somatic mutation of CTNNB1 (#NP_001091679.1) of normal and tumor tissues, spheroids, early and late passages. **d** Correlation matrix between PDS and freshly isolated tumor cells. Early passage for patient 9 corresponds to passage 2. **e** PDS module scores (early and late passages) for the signatures of the five tumor cell types. Source data for signature scores are provided as a Source Data file.

susceptibility, paving a way to use tumor PDS for personalized therapies for this rare tumor.

## Discussion

Hepatoblastoma is a childhood cancer in which the response to chemotherapy can be limited by high-levels of chemotherapeutic resistance[34]. We postulate that an important determinant of the variability in chemosensitivity is the cellular composition of individual tumors, and have utilized the power of single cell transcriptomics to examine shared features of HB that may account for its heterogeneity and relate these features to chemotherapeutic efficacy. We identified five HB tumor signatures that are present in varying proportions across all tumors, supporting the idea that the relative abundance of these tumor cells may underlie the heterogeneity observed in HB. Though

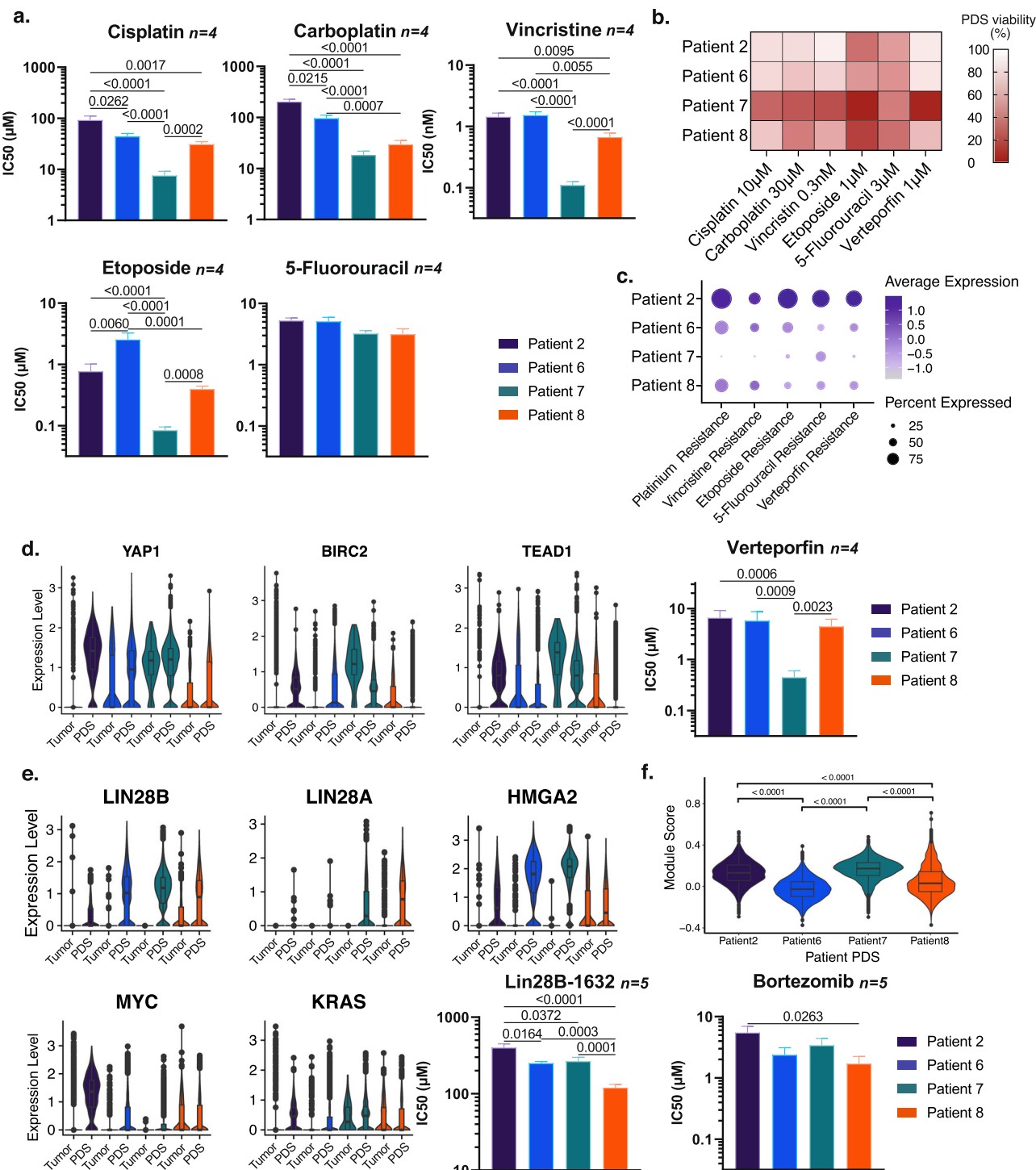

**Fig. 6 | Pharmacologic testing of patient-derived HB spheroids reveals patient-specific treatment responses. a** IC50 calculated from cell viability measurements by ATP quantification after a 4-day treatment with five chemotherapy drugs (cisplatin, carboplatin, etoposide, vincristine and 5-fluorouracil [5-FU]). **b** PDS drug-sensitivity heatmap showing cell viability profile for the five chemotherapy drugs tested and verteporfin at a representative concentration. **c** Module score for genes involved in platinum-based compounds, etoposide, vincristine, 5-FU, and verteporfin metabolism and efflux. **d** Violin plots showing the expression of YAP1, BIRC2 and TEAD1 in parent cells and PDS, and IC50 calculated from cell viability measurements of ATP quantification after a 4-day treatment with the YAP1 inhibitor verteporfin. **e** Violin plots showing the expression of LIN28B, LIN28A, HMGA2, MYC, and KRAS in parent cells and PDS, and IC50 calculated from cell viability measurement by ATP quantification after a 4-day treatment with the LIN28 pathway inhibitor LIN28B-1632. **f** Violin plot showing the module score of proteasome

coding genes (proteasome KEGG gene list) in PDS, and IC50 calculated from cell viability measurement by ATP quantification after a 4-day treatment with the proteasome inhibitor bortezomib. The box plot present the 25th percentile, the median and the 75th percentile, and outlying or extreme values. The whiskers of the box plots extend to a maximum of 1.5 times the size of the interquartile range. IC50 were calculated using the function log(inhibitor) vs. normalized response from GraphPad Prism 9.2.0. Data show log(IC50)+/− SEM generated from drug cytotoxicity assay data (Supplementary Fig. 22) $N = 4$ for cisplatin, Carboplatin, vincristine, etoposide, 5FU and verteporfin, $N = 5$ for LIN28-1632, Bortezomib, Two-way ANOVA was performed, followed by Tukey's multiple comparisons test. p-values are indicated on graphs ($p < 0.0001$ is indicated when p-value is below this threshold). Source data for signature scores and IC50 are provided as a Source Data file.

only five signatures were identified in the current study, our data suggest that additional signatures may need to be defined in order to comprehensively account for the heterogeneity observed in HB tumors. We also identified specific tumor-associated cells, including TAMs, that form the tumor microenvironment. In a subset of HBs, TAMs and tumor cells form a unique niche to maintain fetal liver erythropoiesis. Interestingly, TAMs and tumors express developmentally restricted signals that result in similarly restricted erythropoiesis. Finally, we grew PDS from HB tumors and found that each spheroid culture was an outgrowth of cells enriched in one of the five HB tumor signatures, establishing a useful tool to study HB biology and drug susceptibility. Taken together, this study identifies tumor signatures that may be important for establishing the heterogeneity in this disease and establishes tools to improve our understanding of HB biology and its treatment.

The clinical outcomes for HB depend on the presence of specific epithelial cell types, such as for those patients with small cell undifferentiated histology[35,36]. Consistent with this, bulk transcriptome data separate tumor risk primarily by epithelial component gene expression[8]. Our approach identified two epithelial tumor cell types shared across HB, including cells with pure fetal histology. We also identified three tumor features corresponding to cells in mixed epithelial mesenchymal tumors. This includes a rare neuroendocrine population that correlated with high clinical risk in our dataset. Using this approach on a larger HB dataset could identify most if not all HB tumor cell types. By correlating these with clinical outcomes, it will be possible to define gene signatures that accurately risk stratify HB.

HBs are thought to arise from fetal hepatoblasts, and are known to have few genetic mutations, most in the Wnt signaling pathway, suggesting that tumor heterogeneity in HB is not driven by distinct mutations[37]. In this study, we demonstrate that HBs maintain features that correspond to specific fetal liver developmental stages. This finding supports the hypothesis that HB heterogeneity is determined by the stage at which the tumor arose during liver development[8,11,27].

Patient-derived organoids and spheroids are powerful models to study tumor biology[38,39]. We showed that PDS can be grown efficiently from fresh HB tumor samples and retain patient and tumor cell type gene expression patterns even after long-term culture. Importantly, PDS predicts differential responses to treatment based on the transcriptomic signature of each tumor, suggesting a path forward for precision oncology for these tumors.

Our data is limited by the small sample size of nine tumors, and the fact that eight of our samples are post-treatment, which likely affected the cell types captured and perhaps enriches for chemoresistant tumor cell populations. In addition, it is likely that additional HB tumor signatures exist that we were not able to capture, such as the tumor cells that gave rise to PDS from patient 2. Based on the few number of patients from whom PDS were established and the low number of tumor cells for PDS from patients 2 and 7, additional work will be needed to validate whether drug testing in PDS corresponds to responses in primary tumors. Recent developments in single nuclei RNA sequencing may allow for the examination of larger biobanks of HB samples, which we expect will lead to the identification of other common HB tumor signatures[40].

In conclusion, our results help elucidate the underpinnings of HB tumor heterogeneity with single-cell resolution and demonstrate that PDS can be used to evaluate responses to chemotherapy.

## Methods
### Study approval
The UCSF Institutional Review Board (IRB) committee approved the collection of these patient data included in this study. All relevant ethical regulations for work with human participants have been followed, and written informed consent from all patients' parents or guardians has been obtained.

### Human specimen collection
Fresh tumor tissue and adjacent normal tissue from nine patients were obtained during anatomic liver resections for hepatoblastoma. Tissues were then kept during transportation in ice-cold Williams' E medium (ThermoFisher Scientific, Waltham, MA) (supplemented with 2 mM Glutamax, 10 mM HEPES, and 1000 U/ml Penicillin/Streptomycin (ThermoFisher Scientific)), and then processed for scRNA-seq and spheroid culture.

### Single-cell RNA sequencing
Tumor and adjacent normal liver were dissociated by mincing tissue in 1–2 mm squares, followed by incubation in Liver Perfusion Medium (ThermoFisher Scientific) for 15 min at 37 °C in a rotating oven. After being washed in PBS (ThermoFisher Scientific), tissues were incubated with Liver Digest Medium (ThermoFisher Scientific) supplemented with HEPES and collagenase type IV (Worthington Biochemical, Lakewood, NJ) (600–800 U/ml) for 30 min at 37 °C with rotation. Further dissociation was achieved by pipetting 10 times through a 25 mL serological pipette. Single cells were then separated from clumps using a 70 μm strainer (Fisher Scientific, Hampton, NH). After lysis of red blood cells with ACK RBC Lysis Buffer (Fisher Scientific), single cells were counted using a LUNA™ Automated Cell Counter (Logos biosystems, South Korea) and processed for scRNA-seq analysis.

Sequencing was based on the Seq-Well S^3 protocol[41,42]. One to four arrays were used per sample. Each array was loaded as previously described with approximately 110,000 barcoded mRNA capture beads (ChemGenes, Cat: MACOSKO-2011-10(V+)) and with 10,000–20,000 cells. Arrays were then sealed with functionalized polycarbonate membranes (Sterlitech, Cat: PCT00162X22100) and were incubated at 37 °C for 40 min in lysis buffer (5 M Guanidine Thiocyanate, 1 mM EDTA, 0.5% Sarkosyl, 1% BME). After detachment and removal of the top slides, arrays were rotated at 50 rpm for 20 min. Each array was washed with hybridization buffer (2 M NaCl, 4% PEG8000) and then incubated in hybridization buffer for 40 min. Beads from different arrays were collected separately. Each array was washed ten times with wash buffer (2 M NaCl, 3 mM MgCl₂, 20 mM Tris-HCl pH 8.0, 4% PEG8000) and scraped ten times with a glass slide to collect beads into a conical tube.

For each array, beads were washed with Maxima RT buffer (ThermoFisher, Cat: EP0753) and resuspended in a master mix comprised of Maxima RT buffer, PEG8000, Template Switch Oligo, dNTPs (NEB, Cat: N0447L), RNase inhibitor (Life Technologies, Cat: AM2696) and Maxima H Minus Reverse Transcriptase (ThermoFisher, Cat: EP0753) in water. Samples were rotated first at room temperature for 15 min and then at 52 °C overnight. Beads were washed once with TE-SDS and twice with TE-TW. They were treated with exonuclease I (NEB), rotating for 50 min at 37 °C. Beads were then washed once with TE-SDS, twice with TE-TW, and once with 10 mM Tris-HCl pH 8.0. They were resuspended in 0.1 M NaOH and rotated for 5 min at room temperature. They were subsequently washed with TE-TW and TE, and taken through second strand synthesis with Maxima RT buffer, PEG8000, dNTPs, dN-SMRT oligo and Klenow Exo- (NEB, Cat: M0212L) in water. After rotating at 37 °C for 1 h, beads were washed twice with TE-TW, once with TE and once with water.

KAPA HiFi Hotstart Readymix PCR Kit (Kapa Biosystems, Cat: KK2602) and SMART PCR Primer were used in whole transcriptome amplification (WTA). For each array, beads were distributed among 24 PCR reactions. Following WTA, three pools of eight reactions were made and were then purified using SPRI beads (Beckman Coulter), first at 0.6× and then at a 0.8× volumetric ratio. For each sample, one pool was run on an HSD5000 tape (Agilent, Cat: 5067-5592). The concentration of DNA for each of the three pools was measured via the Qubit dsDNA HS Assay kit (ThermoFisher, Cat: Q33230). Libraries were prepared for each pool, using 800–1000 pg of DNA and the Nextera

XT DNA Library Preparation Kit. They were dual-indexed with N700 and N500 oligonucleotides.

Library products were purified using SPRI beads, first at 0.6× and then at a 1× volumetric ratio. Libraries were then run on an HSD1000 tape (Agilent, Cat: 50675584) to determine the concentration between 100 and 1000 bp. For each library, 3 nM dilutions were prepared. These dilutions were pooled for sequencing on a NovaSeq S4 flow cell.

The sequenced data were preprocessed and aligned using the dropseq workflow on Terra (app.terra.bio). A digital gene expression matrix was generated for each sample, parsed and analyzed following a downstream pipeline.

### Sequencing and alignment

Sequencing results were returned as paired FASTQ reads and the paired FASTQ files were aligned against hg19 reference genome (GRCh37.p13) using the dropseq workflow (https://cumulus.readthedocs.io/en/latest/drop_seq.html). The aligning pipeline output included aligned and corrected bam files, two digital gene expression (DGE) matrix text files (a raw read count matrix and a UMI-collapsed read count matrix where multiple reads that matched the same UMI would be collapsed into one single UMI count) and text-file reports of basic sample qualities such as the number of beads used in the sequencing run, total number of reads, alignment logs. For each sample, the median and average number of genes per barcode were 1014 and 1224. The median and average number of UMI were 2536 and 3974. The mean percentage of mitochondrial content per cell was 17.06%.

### Single-cell clustering analysis

Cells captured in single-cell RNA sequencing analysis were clustered and analyzed using Seurat (Version 3.2) package in R (Version 4.0.3)[43]. Cells with fewer than 300 genes, 500 transcripts, or a mitochondrial level of 20% or greater, were filtered out as the first QC process. Then, by examining the distribution histogram of the number of genes per cell in each sample, we set the upper threshold for the number of genes per cell in each individual sample in order to filter potential doublets. A total of 29,968 cells were acquired using these thresholds.

UMI-collapsed read-count matrices for each cell were loaded in Seurat for analysis. We followed a standard workflow by using the "LogNormalize" method that normalized the gene expression for each cell by the total expression, multiplying by a scale factor 10,000. For downstream analysis to identify different cell types, we then calculated and returned the top 2000 most variably expressed genes among the cells before applying a linear scaling by shifting the expression of each gene in the dataset so that the mean expression across cells was 0 and the variance was 1. This way, the gene expression level could be comparable among different cells and genes. Principal components analysis (PCA) was run using the previously determined most variably expressed genes for linear dimensional reduction and the first 100 principal components (PCs) were stored, which accounted for 40.49% of the total variance. To determine how many PCs to use for the clustering, a JackStraw resampling method was implemented by permutation on a subset of data (1% by default) and rerunning PCA for a total of 100 replications to select the statistically significant PC to include for the K-nearest neighbors clustering. For graph-based clustering, the first 75 PC and a resolution of 1.2 were selected, yielding 37 cell clusters. We eliminated the clustering side effect due to over clustering by constructing a cluster tree of the average expression profile in each cluster and merging clusters together based on their positions in the cluster tree. As a result, we ensured that each cluster would have at least 10 unique DEGs. DEGs in each cluster were identified using the FindAllMarker function within Seurat package and a corresponding p-value was given by the Wilcoxon's test followed by a Bonferroni correction. Two-way ANOVA followed by Tukey's multiple comparisons test was performed using GraphPad Prism version 9.2.0

for Windows, GraphPad Software, San Diego, California USA, www.graphpad.com.

### Cell type signature analysis

In order to annotate each cell type from the previous clustering, we referred to established studies and used signature gene sets for each cell type (Supplementary Data 2). Treating the signature gene set for each cell type as a pseudogene, we evaluated the signature score for each cell in our dataset using the AddModuleScore function. Each cluster in our dataset was assigned with an annotation of its cell type by top signature scores within the cluster. To validate the identities of the tumor cell populations, we estimated copy number variants (CNV) via InferCNV (Version 1.4.0), using non-tumor and non-tumor-associated populations as reference[15]. During the inferCNV run, genes expressed in fewer than five cells were filtered from the data set, and the cut-off was fixed at 0.1. Hidden Markov model (HMM) based CNV prediction was achieved and estimated CNV events were shown in a heatmap.

Somatic copy number alterations were identified and corrected for tumor purity and ploidy. Read counts across 10 kb windows in the tumor and paired normal samples were taken to compute the tumor to paired normal copy number ratio, which can be converted copy numbers.seg files were generated for five HB patients included this study and imported in IGV for visualization.

All tumor cells were subset and re-clustered using the analytical workflow described above. Eight clusters of tumor cells were obtained with distinctive transcriptomic profiles. By downsampling each cluster to 200 or fewer cells, we computed the correlation matrix between each tumor cell pairs and used the pheatmap R package (Version 1.0.12) (Kolde, 2019, https://CRAN.R-project.org/package=pheatmap) to make the correlation heatmap with unsupervised clustering. Gene signatures were tested on all eight tumor cell clusters using previously established hepatoblast, fibroblast, erythroid and neuroendocrine signature gene sets. A one-way ANOVA test was conducted for each signature score comparison across all eight tumor cell clusters, and a corresponding p-value was computed.

In this analysis, we created customized gene set signatures for each cell population of interest. Using the DEGs obtained from FindAllMarker function, we included genes with log2 fold change > 2 and statistical significance (FDR q < 0.05) as the signature gene set.

### RNAscope FISH and immunofluorescence

Immunofluorescence (IF) alone was carried out, on 5 μm-thick FFPE sections, beginning with a deparaffinization, then a 30 min incubation in a steamer in the antigen retrieval CITRA (Biogenex Laboratories, HK086-9K), followed by a blocking step of 1 h in 0.1% Triton + 5% donkey serum. Then primary antibodies, rabbit anti-POSTN (Invitrogen PA534641) at 1:200 and mouse anti-COL1 (Abcam, ab6308) at 1:200, were incubated at 4 °C over the night, and then secondary antibodies, Dylight-755 donkey anti-rabbit IgG (Invitrogen SA5-10043) and Alexa Fluor-467 donkey anti-mouse IgG (Jackson Immuno, 715-605-151), both at 1:1000, were incubated at room temperature for 1 h. All antibodies were diluted in blocking buffer.

RNAscope FISH were carried out on FFPE or Fixed frozen tissue sections of 5 and 7 μm thickness respectively, and were stained with the RNAscope Multiplex Fluorescent Reagent Kit v2 Assay and RNAscope 4-plex Ancillary Kit for Multiplex Fluorescent Reagent Kit v2 (Advanced Cell Diagnostics, Bio-Techne) with Opal 520, 570, 650, 690, 750 and/or 780 (Akoya), following manufacturer instructions. For CHGA immunofluorescence staining following RNAscope experiment, IF protocol was followed starting from blocking step, then with a rabbit anti-CHGA (dilution 1/200, ref: Abcam, ab283265) then a donkey anti-Rabbit Dylight-755. Slides were counterstained and mounted with ProLong™ Gold Antifade with DAPI.

Stained sections were imaged with a Leica DM6B with 0.28 μm z-step size, using a 40× objective and LASX 3.7 software (Leica). Images

were 3D-deconvoluted then an Extended Depth of Focus image was generated using LASX 3.7.

## Tumor-associated erythroid developmental analysis

Tumor-associated erythroid populations were extracted and integrated with the fetal liver erythroid or erythroblast populations from two publicly available datasets (descartes.brotmanbaty.org, human fetal liver erythroblast)[28] (fetal liver early, mid and late erythroid, ArrayExpress, accession code E-MTAB-7407)[14]. We utilized the integration method based on commonly-expressed anchor genes by following the Seurat integration vignette to remove batch effects of samples sequenced with different technologies and possible artifacts so that the cells were comparable.

To evaluate the tumor-associated erythroid and tumor cell populations with respect to the fetal developmental stages, we first calculated a partition-based graph abstraction (PAGA) graph using SCANPY's (Version 1.4.2) (Wolf et al., Genome Biology, 2018) sc.tl.paga() function and then used sc.tl.draw_graph() to generate the PAGA-initialized single-cell embedding of the cell types. The expression of markers was projected from the three fetal erythroid developmental markers (early, mid and late) to each tumor and erythroid cluster to generate a heatmap.

## Pseudotime analysis

The erythroid population was exported as a Seurat object and then converted into a SingleCellExperiment sim object. Pseudotime analysis was conducted using the slingshot R package (Version 1.6.1)[21]. First, PCA decomposition was performed using the prcomp() function in the stats R package (Version 3.6.2). A diffusion map was then generated using the top layer annotation from the original Seurat object and the pseudotime trajectory was superimposed on the diffusion map. The starting point of the pseudotime trajectory was determined based on preliminary understanding of the cell populations used in the analysis.

## Cell-cell interaction analysis

We evaluated cell–cell interactions between two populations of interest using the CellPhoneDB package (Version 2.1.4)[44]. For each analysis, two input files were generated including a normalized gene expression matrix and a two-column metadata for cell names and annotations. The normalized gene expression matrix was obtained by using the NormalizeData function in Seurat with "RC" method specified. Statistical analysis of all available ligand–receptor pairs was performed on local computers.

To investigate the biologically relevant cell populations, we filtered the CellPhoneDB p-value.txt output file for ligand–receptor pairs with the p-value less than 0.05, indicating statistically significant interactions, and generated customized columns and rows txt files. Dot plots were then plotted using these files to illustrate only the significant ligand–receptor interactions.

## Patient-derived HB tumor spheroids

PDS were cultivated either from cryopreserved single cells (patient 2) or from fresh remaining clumps after tissue dissociation (patients 6, 7, 8, and 9) as recently described[45]. Cryopreserved cells were thawed in wash medium (Advanced DMEM/F12 Medium ((ThermoFisher Scientific, Cat: 12634010) containing 2 mM Glutamax (Cat: 35050061), 10 mM HEPES (Cat: 15630080), 1000 U/mL penicillin/streptomycin (Cat: 15140122) and 5% FBS (Fisher Scientific, Cat: 35-016-CV)). Then, cells were centrifuged at $400 \times g$ for 5 min, and resuspended in tumor medium (Advanced DMEM/F12 supplemented with 2 mM Glutamax, 10 mM HEPES, 1000 U/mL penicillin/streptomycin, 2% B27 (Thermo-Fisher Scientific, Cat: 17504044), 1% N2 (ThermoFisher Scientific, Cat: 17502048), 10 mM Nicotinamide (MilliporeSigma, Burlington, MA, Cat: N3376-100G), 1.25 mM N-acetylcysteine (MilliporeSigma, Cat: A9165-5G), 10 μM Y27632 (BioGems, Westlake Village, CA, Cat: 1293823-

10MG), 100 ng/mL hFGF10 (PeproTech, Rocky Hill, NJ, Cat: 100-26), 25 ng/mL hHGF (PeproTech, Cat: 100-39H-25ug), 50 ng/mL hEGF (PeproTech, Cat: AF-100-15-1mg), 5 μM A83-01 (Fisher Scientific, Cat: 29-391-0) and 3 nM dexamethasone (BioGems, Cat: 5000222-5G)). Cells were first seeded in a low binding plate for 4 h at 37 °C, 5% CO$_2$, to promote cell clumping, then clumps were centrifuged, resuspended in pure Matrigel® (Corning, Corning, NY, Cat: 356231) and 25 μL domes were seeded in 48-well plates. After allowing the Matrigel® to solidify at 37 °C, tumor medium was added. For fresh culture, tissue clumps were centrifuged and resuspended in pure Matrigel®, and seeded as previously described. Medium was renewed every 3 to 4 days. Spheroids were visible after 2 to 4 days and passaged after 2 weeks.

Subsequent passaging was performed every 7 to 10 days by splitting cells at a ratio of 1:10 to 1:30. Briefly, domes were washed with PBS, Matrigel® was digested, and cells were dissociated with TrypLE 1X (ThermoFisher Scientific, Cat: 12563011) for PDS from patient 6, 7, 8 and 9 or with TrypLE 10x (ThermoFisher Scientific, Cat: A1217702) for PDS from patient 2, at 37 °C for 20 to 45 min (varying with cell line) until spheroids became small clumps. Then, cells were centrifuged for 5 min at $400 \times g$ at 4 °C and rinsed with wash medium. The cell pellet was then resuspended in pure Matrigel®, and seeded as previously described. Tumor spheroids can be passaged for more than 15 times and about 6 months.

Expression profiling of spheroids was performed by scRNA-seq analysis at an early passage (2–3) and at a late passage (10–11). Briefly cells were prepared as for passaging, using an extended incubation in TrypLE 1X to obtain a single cell suspension. Then, cells were filtered through a 70 μm strainer, counted using a LUNA™ Automated Cell Counter, and loaded into a Seq-Well array as previously described. A total of 15,922 cells passed the QC step from all five cell lines, with an average nGene of 3570 and an average nUMI of 10,264. Signature scores for drug resistance were calculated using the Seurat function AddModuleScore with the following gene lists: Platinum Resistance (*ABCG2, ABCC2, ATP7A, ATP7B, MT1A, MT2A, MPO, GSTP1, GSTT1, GSTM1, SOD1,* and *NQO1*), Etoposide Resistance (*ABCB1, ABCC3, ABCC1, CYP3A5, CYP3A4, UGT1A1, PTGS1, PTGS2, MPO, GSTP1, GSTT1*), Vincristine Resistance (*ABCB1, ABCC10, ABCC1, ABCC2, RALBP1, ABCC3, CYP3A5,* and *CYP3A4*), Verteporfin Resistance (*ABCG2, CES1, CES2, CES3, CES4A,* and *CES5A*), 5-Fluorouracil Resistance (*ABCC3, ABCC4, ABCC5, ABCG2, CDA, CES1, CES2, CYP2A6, DPYD, DPYS, PPAT, RRM1, RRM2, SLC22A7, SLC29A1, TK1, TYMP, TYMS, UCK1, UCK2, UMPS, UPB1, UPP1,* and *UPP2*), DNA Repair (*HMGB1, MLH1, MSH2, MSH6, PMS2, XRCC1, ERCC2, ERCC3, ERCC4, ERCC6, XPA, POLH, POLM, POLB,* and *REV3L*).

Doubling time and drug cytotoxicity were evaluated based on ATP quantification (CellTiter-Glo® 3D Cell Viability Assay kit, Promega, Madison, WI, Cat: G9682). Briefly, cells were seeded at 25,000 cells per dome. For cell growth study, ATP content was measured in domes at days 0, 2, 4, 6, and 8 after seeding. For cytotoxicity drug testing, cells were seeded as previously described and treated with drugs at 4 different concentrations for 4 days, between day 1 and day 5. For each separate experiment, ATP levels were quantified using an ATP-standard curve. Cisplatin (Fisher Scientific, Cat: 232120-50MG) and carboplatin (Sigma-Aldrich, Cat: C2538-100MG) were reconstituted in PBS and all other drugs (vincristine (Sigma-Aldrich, Cat: V0400000), etoposide (EMD Millipore, Cat: 341205-25MG), verteporfin (Sigma-Aldrich, Cat: SML0534-5MG) and 5-fluorouracil (Sigma-Aldrich, Cat: F6627-1G), bortezomib (Sigma-Aldrich, Cat: F6627-1G) and LIN28B-1632 (Fisher Scientific, Cat: 606810)) in DMSO (Cell Signaling Technology, Danvers, MA, Cat: 12611P). Controls were incubated with their respective vehicles (PBS or DMSO).

Data show average +/− SEM of at least 4 independent experiments (see figure legends for details), and correspond to the percentage of ATP levels normalized to the control. IC50 were calculated using the function log(inhibitor) vs. normalized response from GraphPad Prism 9.2.0. Data show log(IC50) +/− SEM. Multiple comparisons among

groups were performed with Two-way ANOVA was performed, followed by Tukey's multiple comparisons test using GraphPad Prism 9.2.0 software. *p*-values are indicated on graphs (<0.0001 is indicated when *p*-value is below this threshold).

### *CTNNB1* mutation detection

DNA was extracted from frozen tissues or cryopreserved freshly isolated cells for tumor and tumor-adjacent samples, and from spheroids at early and late passages, using DNA/RNA All Prep mini Kit (Qiagen, Hilden, Germany, Cat: 80004). *CTNNB1* genomic sequence (from exon 2 trough exon 4) was amplified using specific primers (forward primer: AGCGTGGACAATGGCTACTCAA; reverse primer: ACCTGGTCCTCGTCATTTAGCAGT) by polymerase chain reaction using Q5® Hot Start High-Fidelity 2X Master Mix (NEB, Ipswich, Ma, Cat: M0494S), then sequenced by Sanger sequencing (MCLAB, South San Francisco, CA) using the same primers.

### Reporting summary

Further information on research design is available in the Nature Research Reporting Summary linked to this article.

## Data availability

Fetal liver erythroblasts and hepatocytes publicly available data used in this study are available at descartes.brotmanbaty.org[28]. The fetal liver erythroid publicly available data used in this study are available at ArrayExpress under the accession code E-MTAB-7407[14]. Raw single-cell RNA sequencing FASTQ files and gene expression matrices files generated in this study have been deposited in the Gene Expression Omnibus (GEO) under the accession number GSE186975. The remaining data are available within the Article, Supplementary Information or Source Data file. Source data are provided with this paper.

## Code availability

All software algorithms used for analysis are available for download from public repositories. We have not developed new software in this study. We generated new codes in this study, and the codes used to generate figures in the manuscript will be available in this Github repository [https://github.com/angelussong/Hepatoblastoma_Analysis/].

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

## Acknowledgements

We thank Roel Nusse and Peng Wu for sharing their protocol for HB spheroid cultures, Stuart Orkin for helpful discussions on liver erythroid development, Pam Derish, Xin Chen, and Monty Bissell for their critical review and helpful comments of the manuscript, and the Chan Zuckerberg Biohub for sequencing support. We would also like to thank Alex Shalek for his assistance with Seq-well and the UCSF Liver Center for assistance with cell isolation. This work was supported by the FAVOR NIH T32 Training grant (5T32AI125222-05, DH) American Cancer Society Individual Research Grant (IRG-17-180-1, AN), American Association for the Study of Liver Diseases Clinical, Translational, and Outcomes Research Award (AN), NIH K08DK101603 (BMW), Burroughs Wellcome Fund Career Award for Medical Scientists (BMW), and core resources of the UCSF Liver Center (P30 DK026743).

## Author contributions

H.S. and S.B. equally contributed to experimental design, performing the experiments, data analysis and interpretation, and writing of the manuscript. K.R. performed the experiments and contributed to data analysis and interpretation. M.T. performed the experiments and contributed to data analysis and interpretation. D.B. performed the experiments. D.H. performed the experiments and contributed to data analysis and interpretation. S.J.C. contributed to data analysis and interpretation. A.R. contributed to data analysis and interpretation. M.B., S.L., and A.S.C. performed the experiments and contributed to data analysis and interpretation. M.P. collected samples and obtained patient consents. F.W.H., A.N., and B.W. equally contributed to experimental design, data analysis and interpretation, writing of the manuscript, and supervising the work.

## Competing interests

The authors declare no competing interests.
