## [Peer Review File · Nature Communications]

Single-cell analysis of hepatoblastoma identifies tumor signatures that predict chemotherapy susceptibility using patient-specific tumor spheroidsReviewers' Comments:

Reviewer #1:

Remarks to the Author:

Hanbing and collaborators performed single cell analysis of 9 Hepatoblastomas including 8 after chemotherapy. Histology of the 9 analyzed HB was pure fetal in 5 cases or mixed in the remaining. They identified tumor cell signature related to resistance to chemotherapy.

This is a very interesting set of data. However, analyses could be much more precise.

Single cell analyses of the 9 cases revealed 36 clusters of cells defining 29 different cell types. Among them, 12 clusters were annotated as "tumor cells". Description of the different clusters is confusing. The tumor nature of the cells should be better assessed for example using CNA analyses and CTNNB1 mutations. Also, the transcriptomic profile of each cluster should be compared to the most recent published transcriptomic classifications based on bulk RNAseq of HB (Nagae et al, Nature Com 2021 and Hirsch et al Cancer discovery 2021).

The origin of the 5 different clusters of cancer cells remained to be clearly analyzed according to the corresponding bulk RNAseq and WGS in these patients. Currently the conclusions are too vague.

Analysis of PDS are restricted to a low number of patients and conclusions remained to be validated in primary tumors from patients with follow-up. Once again, signatures should be compared to those previously described as associated to cisplatin resistance (Hirsch et al 2021).

Reviewer #2:

Remarks to the Author:

Hepatoblastoma (HB) is the most common primary pediatric liver cancer, which the response to chemotherapy can be limited by high levels of chemotherapeutic resistance. It's important to dissect the tumor cell signatures that can predict susceptibility to chemotherapy. The single-cell sequencing and spheroids technology provide the opportunity to predict treatment response and identify novel therapeutic targets in a novel method. In this manuscript, the authors performed an integrative analysis of single-cell transcriptomic profiling to define HB tumor heterogeneity with single-cell resolution and demonstrated that patient-derived spheroids can be used to assess chemotherapy response. In general, this work provided unique resources, and the quality of the data generated and the individual analysis performed is good. This work is potential of broad interest to the cancer therapeutic community for HB tumors in that the authors found tumor heterogeneity and drug sensitivity of chemotherapy on HB patients may be in a patient-specific manner.

This article focuses on discovering the relationship between HB tumor characteristics and chemotherapy response. However, the author defines that tumor characteristics have strong sample specificity. The author needs to determine whether the tumor characteristics are sample-specific or caused by batch effects in the analysis. Then, the author needs to give a reasonable explanation for how this tumor feature itself affects the patient's prognosis, and how it is related to chemotherapy sensitivity. The authors further need to address the following concerns.

Other major comments:

1. The author claimed that their data showed an inter-sample batch effect. However, in supplementary figure 3b, the different patients were assigned in different clusters. The authors need to compare the results between batch correction and without batch correction to make their results convincing. In addition, tumor cell cluster 1-5 only includes one patient in each cluster in figure 2b, which may suggest the high heterogeneity or batch effect for tumor cells among the patients. The authors need to validate whether this different grouping is caused by heterogeneity or batch effect.
2. There are several cell clusters that express HB markers. Are these caused by experimental doublets, or are they really existing clusters? Authors need to verify their conclusions and explain why

these cell types express HB markers.

3. On page 8, the authors described that “the HB-associated erythroid cells separately and showed that cells from Patients 2 and 5 were predominantly early erythroid, whereas those from Patient 6 were primarily mid to late-stage erythroid cells”. Do this phenomenon associated with the patient's age? In addition, why do only patients 2, 5, and 6 have HB-associated erythroid cells? Is it possible from blood or erythroid cell contamination in the preparation of single-cell suspension?

4. On pages 11-12, the authors described that “we did not capture the parent cells for PDS from Patient 2 and very few tumor cells were captured from Patient 7”. In figure 6, they examined the drug sensitivity of these PDSs and found that “spheroids from Patient 7 were the most sensitive to the drugs tested, while PDS from Patients 2 were the most resistant”. The drug sensitivity of these spheroids may not reflect the actual drug response of the corresponding patient.

Minor comments:

1. The description order of the figures in the manuscript is not according to figure order, like Figure S3 before Figure S2 and Figure S4 before Figure S3 on page 4.

2. What cell type did the author use as reference cells in Figure 1e? What cell types do tumor-associated cells include in Figure 1e?

3. On page 8 lines 6-7, the authors mentioned that NMB, FOSL2 enriched in MACROlow macrophage related to figure 3h and i, but the corresponding figures don't show these two genes.

4. In figure 3k, the authors displayed ligand-receptor interactions between MARCOlow TAMs and four of the five HB tumor cell types. They should compare these results with MARCOhigh TAM and HB tumor cell types to confirm that these interactions are only present in MARCOlow TAM.

5. The author claimed that “We ensured that the presence of erythroid genes in these cells was not due to ambient RNA contamination (Supplemental Figure 13) or doublets (data not shown).” on page 10. But the reviewers could not find which panel in Figure S13 explained the results they claimed. Again, authors need to label in the context which panel they describe corresponds to in the supplementary figure. It is very confusing when they only mark the entire figure.

6. In figure 6d, please list “genes involved in platinum-based compounds, etoposide, vincristine, 5-FU, and verteporfin metabolism and efflux” in the method.

7. In figure 6f, BRCA1, and FANC are expressed in all PDSs but low or not expressed in patient tumors. It needs more evidence to explain why patient 8 cells were sensitive to the inhibitor of the BRCA/FANC pathway.

Reviewer #3:

Remarks to the Author:

In the manuscript entitled “Single-cell analysis of hepatoblastoma identifies distinct tumour cell signatures that predict susceptibility to chemotherapy using patient-specific tumour spheroids” the authors use single cell RNA-sequencing (scRNA-seq) with the aim to investigate tumour heterogeneity. The authors claim to have identified five tumour cell subtypes which can be used to define tumour heterogeneity and predict chemotherapy/ treatment response.

Paediatric cancers are, compared to those of adults, understudied and the clinicopathological diversity of HB is just starting to be uncovered. Thus, research providing insights into HB pathology and treatment is of great interest. The manuscript does, unfortunately, not significantly contribute to this end. This is mainly due to the fact that most differences are caused by the fact that different tumour subtypes are being compared to each other. The existence of different tumour subtypes is known and there is no comparison of previous subtype classifications to the signatures made by the authors. The tumour spheroids could be a valuable model system but needs further characterization.

The dataset could still be valuable and if re-analysed could have the potential to provide valuable information about the disease. Below we list points and suggestions that we think can improve the manuscript.

Major points

- Abstract row 6: transcriptomic not genomic techniques. Although first strand synthesis and amplification are performed during library preparation what is being analysed is the transcriptome and not the genome. This is an important difference.
- The cut-off values of >300 genes/cell and > 500 transcripts/ cell are low. Common values are >500-800 genes/cell and ~2000 transcripts/cell for very high throughput scRNA-seq methods such as 10x (even higher for more sensitive methods such as SMART-Seq or CEL-Seq2)^{1,2}. Could the authors please comment on their choice of cut-off values. Based on Supp fig1 higher cut-off values could have been used.
- Supp fig2b show that the number of cells from patient 3, 7, 9 are very low. Especially since these cells are divided over multiple clusters. Could the authors please comment on the number of cells they consider necessary to draw statistically significant conclusions and to what degree they believe these cells are representative of the tumour and its microenvironment.
- When identifying tumour cell types the authors seem to include both epithelial and non-epithelial cells? Is this correct, and could the authors please comment on the rationale behind this. Differences are expected when comparing a tumour containing a lot of fibrotic tissue to one that does not. To discover less obvious differences comparing cell types present in all the tumours could be performed.
- The authors claim that "This study establishes that tumor heterogeneity can be defined by the relative proportions of five distinct subtypes of tumor cells." To ensure that the identification of tumour cell subtypes (present in all tumours) excludes patient-to-patient variability the authors should provide proof that there are minimal differences between the matched healthy cells from all the patients. Clustering and UMAP of showing the patient identity of the cells is needed.
- In the beginning of the manuscript the authors write tumour cell subtype and later on, the authors are identifying 5 tumour signatures. This makes the reader wonder whether the aim of the authors is to establish tumour signatures that can contain multiple different cell types or to identify 5 individual tumour cell types? This needs to be clarified throughout the manuscript as the authors are trying to identify signatures not cell types.
- The authors subcluster only the tumour cells and identify 8 clusters which are then divided into five distinct groups – forming the five different signatures. Some signatures contain tumour cell clusters from a subset of patients. This conflicts with the authors statement that "We identified five transcriptomically distinct tumor cell signatures that are present in varying proportions across all tumors, indicating that the relative abundance of these tumor cells underlies the heterogeneity observed in HB." Could the authors please comment.
- The authors compare their data to bulk tissue gene expression datasets. However, as there is a scRNA-Seq dataset available³ it would be more relevant to see how the authors data compare to this.
- For the MARCO ligand-receptor pairs, transcriptome data is suggestive of interaction but not definitive proof. If the authors want to claim interaction stainings or other validation is necessary.
- Validation of the five tumour cell types by immunohistochemistry/immunofluorescence or in situ RNA hybridisation would greatly improve the quality of the manuscript.
- How do the spheroids compare to healthy tissue? In some cases, healthy tissue can take over or 'contaminate' the culture if the tumours are slow growing. Can be one of the reasons for the relatively low correlation between spheroid and tumour tissue in fig 5d
- More data regarding the spheroids would improve the manuscript.
- Chromosomal status
- How often they can be passaged?
- Brightfield images of late and early passages to show the consistent morphology and growth of the cultures
- Histological analysis such as H&E of the spheroids to show that they recapitulate the primary tumour tissue
- The spheroids have CTNNB1 mutations. Do the authors see an increased expression of Wnt target genes?
- The authors should show sanger sequencing traces of both the primary tumour tissues and spheroids confirming the CTNNB1 mutations

- Did the authors establish hepatocyte organoids from the healthy tissue? Being able to perform drug screens on tumour spheroids and matched healthy organoids would be a valuable addition to the manuscript.

Minor points

- Add the following info to the table summarising the patient information: whether there is matched normal tissue or not, the total number of cells and the number of cells passing quality control (can still keep the bar graph).
- “One of the nine patients had tumor extension into the adjacent tissue and therefore non-tumor tissue was unavailable.” – state which patient in the main text.
- For all the violin plots do box plots within the violin plots rather than single dots, as it is not possible to see the shape of the violin plots due to the high number of dots.
- Fig 1f. Does the patient contribution to each cluster reflect the proportion of cells (relative to the number of cells from the other patients) coming from that patient? Could the authors please comment.
- Fig 1b. do a legend with the text rather than writing it out in the UMAP. It is difficult to read.
- Fig2c Could the authors please draw dotted lines in the correlation heatmap to mark the different signatures and include a table summarising which tumour cell clusters are included in each signature?
- Fig 3k Figure legend correction -Not for each patient, but for each tumour signature. By including the MARCOHigh cells in the dot plot the difference can be seen more easily.
- Show UMAP plots of some of the representative genes for each tumour signature.
- With regards to the authors analysis of erythroid cells, have the authors tried to add some of the factors secreted by these cells to their spheroid cultures to see if it affects their growth?
- PCA plots showing how tumour and matched normal samples cluster relative to each other.
- Stainings of marker genes of the spheroid cultures
- Supp fig 7. Use dotted lines to indicate tumour tissue.
- Supp fig 9. Please also do the sub clustering of the healthy tissue and the healthy and tumour tissue together
- Supp fig 14b growth curves would be more informative

References

1. Elmentaite, R. et al. Cells of the human intestinal tract mapped across space and time. *Nature* 597, 250–255 (2021).
2. Haber, A. L. et al. A single-cell survey of the small intestinal epithelium. *Nature* 551, 333–339 (2017).
3. Nagae, G. et al. Genetic and epigenetic basis of hepatoblastoma diversity. *Nat. Commun.* 12, 5423 (2021).

REVIEWER COMMENTS

Reviewer #1, expertise in paediatric hepatoblastoma genomics and transcriptomics

1. Single cell analyses of the 9 cases revealed 36 clusters of cells defining 29 different cell types. Among them, 12 clusters were annotated as “tumor cells”. Description of the different

clusters is confusing. The tumor nature of the cells should be better assessed for example using CNA analyses and CTNNB1 mutations.

We thank the reviewer for this comment and apologize for the confusion caused by our current description of the different clusters identified in our analysis. We have made the following edits in the manuscript to clearly describe the 12 clusters that were composed of cells that predominantly originated from tumor specimens as either *tumor clusters* (6) or *tumor-associated clusters* (6). *Tumor clusters* were identified based on high expression of known HB tumor markers including *REG3A*, *MEG2*, and *IGF2*. *Tumor associated clusters* were denoted as those clusters for which >60% of the cells originated from the tumor without expression of HB tumor markers. These edits have been made throughout the manuscript and we hope this will clarify any confusion.

We also agree that CNA and mutational analysis of CTNNB1 would be of benefit. For this reason we have included the mutational analysis of our 9 patients as a new supplementary figure (Supplementary Fig. 7). Seven of our nine patients have specific mutations in CTNNB1 that are present in each tumor sample and absent in adjacent normal liver. Notably, these mutations are also conserved in the respective PDS. We have also performed a CNA analysis for 5 of the 9 tumors and have found patient specific differences in the copy number pattern (Supplementary Fig. 7b). Whole genome sequencing of paired normal and tumor tissue will be needed to define CNA at a higher resolution and was not available for the current study. Along with the inferCNV analysis demonstrating that the 6 tumor clusters exhibited copy number variations (Fig. 1e), these data demonstrate the tumor nature of the cells included in tumor clusters. These data have been included in the Results of the revised manuscript (page 5, line 13).

2. Also, the transcriptomic profile of each cluster should be compared to the most recent published transcriptomic classifications based on bulk RNAseq of HB (Nagae et al, Nature Com 2021 and Hirsch et al Cancer discovery 2021).

We sincerely appreciate this comment as the papers cited do offer helpful bulk RNA sequencing data. We have compared published bulk RNAseq datasets to our individual tumor cell clusters by extracting gene sets of each published transcriptomic classification from the two HB bulk RNAseq studies, as this reviewer suggested. The Hirsch et al.² study classified HB into four transcriptomic classifications: *proliferation*, *mesenchymal*, *progenitor*, and *hepatic differentiation*; while the Nagae et al.¹ study categorized HB into three classifications: *mesenchymal*, *hepatocyte* and *proliferative*.

We computed the transcriptomic score of each published classification and determined which patients in our dataset had enrichment of these published HB classifications. Patients 1, 4, 6, and 7 showed significant enrichment of the *hepatic differentiation* classification described by Hirsch et al.², and the *hepatocyte* classification reported by Nagae et al.¹ These data are consistent with our analysis as these four tumors were epithelial subtypes of HB. Despite lower mean expression for the *hepatocyte* classification described by Nagae et al.¹, outlier cells from patients 2, 5, and 8 had high *hepatocyte* classification expression (Supplementary Fig. 12e), supporting our conclusion that multiple tumor signatures are present in each patient.

Nagae et al. “Hepatocyte” & Hirsch et al. “Hepatic Differentiation” classification

Analysis of our mixed HB subtypes were also consistent with bulk RNA seq data reported by Hirsch et al.² and Nagae et al.¹, as the patients with mesenchymal components (patients 2, 3, and 5) had the highest expression of the *mesenchymal* classification (Supplementary Fig. 12e).

Nagae et al. “Mesenchymal” & Hirsch et al. “Mesenchymal” classification

We also examined the *proliferative* classification from Nagae et al.¹ Patient 8 showed a significant enrichment for this classification compared to other patients, consistent with the detection of a MKI67+ tumor subcluster within patient 8 (Supplementary Fig. 12e). Since the majority of patient 8 tumor cells expressed the neuroendocrine markers, *CHGA*+ or *CHGB*+, we classified patient 8 as a neuroendocrine type HB.

Nagae et al. “Proliferative” & Hirsch et al. “Proliferation” classification

Finally, we tested the *progenitor* classification from Hirsch et al.², and found that only patient 6 showed enrichment of this classification. One potential explanation for this finding is that due to the high degree of tumor heterogeneity in patient 6, only a subset of patient 6 tumor cells overlap with the *progenitor* classification while the remainder of tumor cells overlap with the *hepatocyte* classification.

Hirsch et al. “Progenitor” classification

We also compared the bulk signatures in these two studies with the five HB signatures we identified in our analysis. Both the Nagae et al.¹ *hepatocyte* and Hirsch et al.² *hepatic differentiation* classifications showed strong association with our hepatoblast I and hepatoblast II signatures. The *proliferative/proliferation* classification partially overlapped with the neuroendocrine HB signature. And the two HB subtypes with mesenchymal components (erythroid-like and DCN-high) exhibit enrichment of *mesenchymal classification* (Supplementary Fig. 12f). These data demonstrate that there is overlap between the transcriptomic signatures reported in our study and the transcriptomic classifications that have been previously reported using bulk RNAseq datasets. The results of our comparison with existing bulk RNA datasets have been included in a revised Supplementary Figure 12 and in the Results (page 7, line 11).

In summary, our data is consistent with previously published bulk RNAseq data on human HB as the tumors that display epithelial and mixed epithelial/mesenchymal gene expression have the highest expression of the respective transcriptomic classifications reported by Hirsch et al.² and Nagae et al.¹ Though we can identify tumor cells that express the *proliferative* and *progenitor* classification, our analysis found that these classifications do not accurately describe the predominant gene expression profile of the cells in our analysis. Taken together, the published classifications from bulk RNAseq datasets do not capture all the transcriptomic heterogeneity we identify using scRNAseq analysis, leading us to propose the five HB signatures in the present study.

3. The origin of the 5 different clusters of cancer cells remained to be clearly analyzed according to the corresponding bulk RNAseq and WGS in these patients. Currently the conclusions are too vague.

We thank this reviewer for this comment and have investigated the origins of the 5 clusters of tumor cells by examining their association with known HB-tumor (e.g. *REG3A*, *MEG2*, and *IGF2*) and liver specific genes (e.g. *DLK1*). In the overall UMAP, we identified six tumor clusters and we observed shared transcriptomic profiles among patients 1, 4, and 6. Then by sub-cluster analysis of the tumor cells, we grouped sub-clusters together based on their gene expression profiles and identified the 5 different HB signatures. As mentioned in the preceding points, we have compared our 5 HB signatures with existing bulk RNAseq datasets along with CTNNB1 mutation and CNA analysis, supporting the identity of the cells within our tumor clusters as actual tumor cells. Taken together, we are confident that the five HB tumor signatures we have identified are composed of HB tumor cells. We hope this additional analysis satisfies this reviewers' concerns.

4. Analysis of PDS are restricted to a low number of patients and conclusions remained to be validated in primary tumors from patients with follow-up. Once again, signatures should be compared to those previously described as associated to cisplatin resistance (Hirsch et al 2021).

We thank the reviewers for this comment and agree that due to the limited number of samples in our analysis, we have a low number of PDS. We also agree with this reviewer's comment that future studies will be needed to validate our observations from primary tumors. We have included this limitation in the discussion section of the revised manuscript (page 16, line 24). This study does, however, highlight the utility of HB PDS cultures to evaluate sensitivity to chemotherapeutics for this rare tumor.

We have compared the signatures associated with cisplatin resistance from Hirsch et al.² and Galluzzi et al.⁴ with the gene expression from our PDS. Hirsch et al.², described two signatures that are associated with cisplatin resistance. One of the signatures is comprised of a list of 63 genes originally described by Galluzzi et al.⁴ We calculated a signature score using this 63-gene signature and found that similar to our initial findings of cisplatin resistance, PDS from patient 7 had the lowest score (low cisplatin resistance), while PDS from patient 2 had the highest score (high cisplatin resistance). Moreover, 8 out of 12 genes we originally used to calculate a cisplatin resistance score were included in Galluzzi et al.'s 63-gene signature.⁴

The additional cisplatin gene signature described by Hirsch et al. (SBS35 signature) is based on a set of mutations induced by cisplatin.² Though we agree with this reviewer that mutation analysis of PDS would be an informative analysis, WGS would be required and could not be completed due to time and cost for the current study.

Reviewer #2, expertise in single cell RNA-sequencing

1. The author needs to determine whether the tumor characteristics are sample-specific or caused by batch effects in the analysis.

We thank the reviewer for this comment. In the present study, we did not perform any batch effect correction because these samples were processed using the exact same protocol and sequenced using the same sequencer. We also observed that stromal and immune cells from different patients clustered together according to their transcriptomic profiles, supporting our conclusion that differences in tumor characteristics are not caused by batch effects, and reflect true distinctions between tumor cells.

2. Then the author needs to give a reasonable explanation for how this tumor feature itself affects the patient's prognosis, and how it is related to chemotherapy sensitivity.

We thank this reviewer for this comment. We acknowledge that a limitation of this study is the limited number of samples available for analysis. Additional HB tumors would be needed to correlate the transcriptomic signatures described in our study with a patient's prognosis. We also acknowledge that no patients had small cell undifferentiated histology and therefore no patients in our study had a well-established subtype of HB with a known survival disadvantage. Despite these limitations, we have identified differential chemotherapy sensitivity based on resistance scores to platinum, vincristine, etoposide, 5-FU, and verteporfin. Though these data support a correlation between a specific HB signature and chemotherapy sensitivity and possibly patient outcome, further research will be needed to validate these findings with a larger set of HB tumor samples.

Other major comments:

3. The author claimed that their data showed an inter-sample batch effect. However, in supplementary figure 3b, the different patients were assigned in different clusters. The authors need to compare the results between batch correction and without batch correction to make their results convincing. In addition, tumor cell cluster 1-5 only includes one patient in each cluster in figure 2b, which may suggest the high heterogeneity or batch effect for tumor cells among the patients. The authors need to validate whether this different grouping is caused by heterogeneity or batch effect.

We thank the reviewer for this comment. In the present study, we did not perform any batch effect correction because these samples were processed using the exact same protocol and sequenced using the same sequencer. We observed that stromal and immune populations clustered together according to the transcriptomic profiles, and not by patient. Therefore, we merged the samples together for our analysis.

We also agree with the reviewer that in order to complete a more thorough analysis of our tumor cell characterization, we need to ensure that our tumor cell identification was not due to batch effect. To address this, we performed an integration among the nine patients using the Seurat integration vignette using individual datasets from each patient. Then using the integrated object, we were able to perform clustering. When we overlap our cluster annotation from the original merged object (Fig. 1b) with this integrated object (see UMAP below), we observe a similar pattern of stromal and immune cells that cannot be separated by patients and that the tumor cell clusters remain distinct (see barchart below), indicating that our annotation captures the heterogeneity of tumor cells between patients instead of batch effect.

4. There are several cell clusters that express HB markers. Are these caused by experimental doublets, or are they really existing clusters? Authors need to verify their conclusions and explain why these cell types express HB markers.

We thank the reviewer for this valid concern. When we used DecontX to decontaminate the ambient RNA and DoubletFinder to locate potential doublets, we did not detect any tumor cluster or tumor associated clusters that are comprised of doublets, nor did we identify clusters with HB markers that also show high-levels of ambient RNA contamination.

We acknowledge that some genes are easier to capture than others in scRNAseq analysis which could explain why the expression of certain HB markers are also present in non-tumor clusters. To account for this possibility, we based our annotation on a set of genes instead of a single tumor marker, to minimize the possibility that a few number of genes will define a tumor cluster. Furthermore, with this strategy, doublets are unlikely to lead to the enrichment of an overall HB tumor signature. We also computed the percentages of doublets in each tumor cell cluster using DoubletFinder, and found that only 3.7% (223 of 6,021 cells) of all tumor cells were labeled as possible doublets, supporting our conclusion that doublets have a limited effect on the overall transcriptomic signatures of tumor cells.

5. On page 8, the authors described that “the HB-associated erythroid cells separately and showed that cells from Patients 2 and 5 were predominantly early erythroid, whereas those from Patient 6 were primarily mid to late-stage erythroid cells”. Do this phenomenon associated with the patient's age?

We thank this reviewer for this insightful point. We did not find an association with the age of a patient and the developmental stage of erythroid cells found in a patient's tumor. For example, the patients with early erythroid cells (patients 2 and 5) were 37 and 30 months, respectively. Patient 6 who had mid to late-stage erythroid cells was 32 months. Notably, due to the developmental origins of HB, all three of these patients were within a limited age range of only 5 months.

6. *In addition, why do only patients 2, 5, and 6 have HB-associated erythroid cells? Is it possible from blood or erythroid cell contamination in the preparation of single-cell suspension?*

We appreciate this comment. Though our methods for single cell isolation include red blood cell lysis, we have performed a more detailed analysis of the erythroid cells found in HB tumors to address this Reviewer's comments. The patients with HB-associated erythroid cells have nucleated erythroid cells, indicating that these are distinct from the non-nucleated erythrocytes one would expect to find from circulating red blood cell contaminants. These data argue against the possibility that our findings are related to blood or erythroid cell contamination. Furthermore, our data is supported by the known pathology of HB as extramedullary erythropoiesis can be a characteristic feature of HB tumors.⁵

7. *On pages 11-12, the authors described that "we did not capture the parent cells for PDS from Patient 2 and very few tumor cells were captured from Patient 7". In figure 6, they examined the drug sensitivity of these PDSs and found that "spheroids from Patient 7 were the most sensitive to the drugs tested, while PDS from Patients 2 were the most resistant". The drug sensitivity of these spheroids may not reflect the actual drug response of the corresponding patient.*

We appreciate this comment and agree that our findings re PDS drug susceptibility in patients 2 and 7 needs to be qualified based on the incomplete capture of these patient's parent cells. We have included this limitation in the discussion (page 16, line 24).

Minor comments:

1. *The description order of the figures in the manuscript is not according to figure order, like Figure S3 before Figure S2 and Figure S4 before Figure S3 on page 4.*

We apologize for this oversight and have corrected the figure order in the manuscript text. Specifically, with regard to the example this reviewer has included, Supplementary Figures 1-3 are listed in order in paragraph 1 of page 4. Also, since we have added new supplementary figures based on the reviewers' comments, the numbering of our supplementary figures has changed.

2. *What cell type did the author use as reference cells in Figure 1e? What cell types do tumor-associated cells include in Figure 1e?*

The reference cells used for Figure 1e were the 17 clusters of non-tumor associated cells. The cell types included as tumor associated cells include the following 6 tumor-associated clusters: Pro-myelocyte, basophils, macrophage, WNT5A int fibroblast, WNT5A high fibroblast, and erythroid.

3. *On page 8 lines 6-7, the authors mentioned that NMB, FOSL2 enriched in MACROlow macrophage related to figure 3h and i, but the corresponding figures don't show these two genes.*

We thank the reviewer for this comment and appreciate that our presentation of these data led to the inconsistency highlighted by this reviewer. We have clarified Glycoprotein NMB instead as glycoprotein nonmetastatic melanoma protein B (GPNMB) and this gene is included in Fig. 3e,h,i (page 9, line 23). FOSL2 should not have been included in the main text of the manuscript and has now been removed accordingly. We hope these clarifications address this reviewers' comments.

4. *In figure 3k, the authors displayed ligand-receptor interactions between MARCOlow TAMs*

and four of the five HB tumor cell types. They should compare these results with MARCO^{high} TAM and HB tumor cell types to confirm that these interactions are only present in MARCO^{low} TAM.

We appreciate this point and have performed our ligand-receptor interactions to identify those interactions that are specific to MARCO^{Low} TAMs and the 5 HB signatures. We have decided to move this analysis to supplementary data (Supplementary Fig. 14) as attempts to validate these interactions based on Reviewer #3's comments using spatial transcriptomics have been technically challenging.

5. The author claimed that “We ensured that the presence of erythroid genes in these cells was not due to ambient RNA contamination (Supplementary Figure 13) or doublets (data not shown).” on page 10. But the reviewers could not find which panel in Figure S13 explained the results they claimed. Again, authors need to label in the context which panel they describe corresponds to in the supplementary figure. It is very confusing when they only mark the entire figure.

We appreciate this reviewer's point. In order to avoid ambient RNA contamination confounding our analysis, we intentionally examined erythroid marker genes after samples underwent ambient RNA decontamination using DecontX. DoubletFinder was also used to identify possible doublets and we found that, at most, 3.7% of all tumor cells are potential doublets. We had

mistakenly referred to Supplementary Fig. 13 in the original manuscript and have removed this citation as the use of DecontX and DoubletFinder was part of our analytic pipeline. We have also referred to specific figure panels throughout the entire manuscript to avoid confusion.

6. In figure 6d, please list “genes involved in platinum-based compounds, etoposide, vincristine, 5-FU, and verteporfin metabolism and efflux” in the method.

We thank the reviewer for this comment and agree that we should have provided a list of genes involved in platinum-based compounds, etoposide, vincristine, 5-FU, and verteporfin metabolism and efflux. These genes have been included in the Methods section of the manuscript. They also correspond to the genes listed in Supplementary Fig. 21a-f.

7. In figure 6f, BRCA1, and FANC are expressed in all PDSs but low or not expressed in patient tumors. It needs more evidence to explain why patient 8 cells were sensitive to the inhibitor of the BRCA/FANC pathway.

We thank this reviewer for this comment and agree that the expression of BRCA1 and FANC does not adequately explain the sensitivity to Bortezomib that we observe in PDS for patients 6 and 8. Given the known mechanism of action of Bortezomib as a proteasome inhibitor, we performed additional analysis of proteasome genes among the 4 PDS in our study. Lower expression of proteasome genes has been correlated with greater susceptibility to Bortezomib.^{6,7} Among our PDS, we found that PDS from patients 6 and 8 had lower expression of proteasome genes compared to PDS from patients 2 and 8 (Fig. 6f). These data indicate that lower proteasome genes among PDS from patients 6 and 8 was associated with the greater susceptibility we observe among PDS from patients 6 and 8 to Bortezomib.

Reviewer #3, expertise in patient derived organoids

Major points

1. Abstract row 6: transcriptomic not genomic techniques. Although first strand synthesis and amplification are performed during library preparation what is being analysed is the transcriptome and not the genome. This is an important difference.

We thank this reviewer for this valid point and apologize for this oversight. We have corrected the abstract accordingly (page 2, line 6).

2. The cut-off values of >300 genes/cell and > 500 transcripts/ cell are low. Common values are >500-800 genes/cell and ~2000 transcripts/cell for very high throughput scRNA-seq methods such as 10x (even higher for more sensitive methods such as SMART-Seq or CEL-Seq2)^{1,2}. Could the authors please comment on their choice of cut-off values. Based on Supplementary Fig.1 higher cut-off values could have been used.

We thank the reviewer for this comment. While we acknowledge that using the Seq-well platform for scRNA-seq might not yield as many high-quality cells as 10x, even with a higher QC threshold, we are able to capture the same cell populations as we would with a lower QC

threshold. In other words, we identified the same clusters with high and low QC thresholds (Supplementary Fig. 2b). No cell cluster was eliminated after increasing our QC cut off from 300 genes, 500 UMIs to 500 genes, 1000UMIs (Supplementary Fig. 2c,d). However, to fully appreciate the transcriptomic profiles of each cell cluster, especially the tumor cells and HB associated cell populations, we require as many cells as possible. Therefore, in our analysis, we kept the QC cutoff as 300 genes and 500 UMIs. We also generated a dot plot for the highQC cutoffs using the top differentially genes we identified with the lowQC cutoff (Supplementary Fig. 2e). The dot plot using the highQC threshold is almost identical with the one for the lowQC cutoff, confirming the validity of our profiling of the cell populations we identified. We have included this information and referenced Supplementary Fig. 2 on page 4, line 5.

3. Supplementary Fig.2b show that the number of cells from patient 3, 7, 9 are very low. Especially since these cells are divided over multiple clusters. Could the authors please comment on the number of cells they consider necessary to draw statistically significant conclusions and to what degree they believe these cells are representative of the tumour and its microenvironment.

We appreciate this reviewer's comments and agree that there are fewer cells captured in patients 3, 7, and 9. Despite the relatively few number of tumor cells in patients 7 and 9, we found that tumor cells in these patients still exhibited significant enrichment in the Hepatoblast I signature, which is consistent with the histological classification of these two patients. For this reason, we believe that the tumor cells captured from patients 7 and 9 are representative of the predominant tumor signature. We acknowledge that in order to identify other tumor signatures present in these patients, we would need at least 200-300 cells for each patient. This estimate is based on the need for >50 tumor cells per tumor cell subcluster and the identification of at least 5 tumor signatures. We did capture 1,210 tumor cells from patient 3, which was sufficient to analyze their transcriptomic profiles. Based on our validation of tumor cells using CTNNB1 mutation analysis, CNA, and inferCNV, we are confident that the cells captured are in fact tumor cells. We also noticed that all the non-parenchymal cells identified from each tumor (immune, stromal, endothelial cells) had contributions from every patient including those that had relatively lower number of cells captured. These data support the idea that tumor microenvironment is accurately represented by the cells captured.

4. When identifying tumour cell types the authors seem to include both epithelial and non-epithelial cells? Is this correct, and could the authors please comment on the rationale behind this. Differences are expected when comparing a tumour containing a lot of fibrotic tissue to one that does not. To discover less obvious differences comparing cell types present in all the tumours could be performed.

Our approach to this analysis was to be agnostic to the cell types that we were analyzing in the HB tumor specimens. We also wanted to prioritize capturing all the cells in these rare tumors as it remains unclear which cell(s) drive the histological diversity and account for HB biology and clinical outcomes. In our opinion, our analysis provides an in-depth characterization of the tumor cells that contribute to the histological diversity observed in HB.

With regards to this reviewer's second point re comparing cell types present in all tumors, we have analyzed cell types that are conserved across all tumors (immune cells) and have found few differences in the transcriptomes of these cells across patients (Supplementary Fig. 14). This supports the idea that the heterogeneity of these tumors is dependent on a subset of cells found within tumors, namely the five HB signatures that we have identified.

5. The authors claim that "This study establishes that tumor heterogeneity can be defined by the relative proportions of five distinct subtypes of tumor cells." To ensure that the identification of tumour cell subtypes (present in all tumours) excludes patient-to-patient variability the authors should provide proof that there are minimal differences between the matched healthy cells from all the patients. Clustering and UMAP of showing the patient identity of the cells is needed.

The reviewer raises an excellent point. We generated the UMAP below by showing the distribution from the nine patients. The clusters derived from non-tumor populations (stromal, immune populations, etc.) are composed of multiple patients supporting the idea that patient to patient variability drives transcriptomic differences of tumor cells between patients. These data have been included in Supplementary Fig. 4b.

6. In the beginning of the manuscript the authors write tumour cell subtype and later on, the authors are identifying 5 tumour signatures. This makes the reader wonder whether the aim of the authors is to establish tumour signatures that can contain multiple different cell types or to identify 5 individual tumour cell types? This needs to be clarified throughout the manuscript as the authors are trying to identify signatures not cell types.

We thank the reviewer for this comment and apologize for any confusion in our manuscript. The primary focus of this study was to define tumor heterogeneity of HB by identifying specific tumor signatures that are present, to varying degrees, within different patients. In order to identify signatures, we first identified tumor cells and noted these cells clustered in 6 groups (i.e. 6 tumor clusters). Further sub-clustering analysis of these 6 tumor clusters identified five HB signatures. We have clarified this important point throughout the manuscript and hope these revisions alleviate any confusion.

7. The authors subcluster only the tumour cells and identify 8 clusters which are then divided into five distinct groups – forming the five different signatures. Some signatures contain tumour cell clusters from a subset of patients. This conflicts with the authors statement that “We identified five transcriptomically distinct tumor cell signatures that are present in varying proportions across all tumors, indicating that the relative abundance of these tumor cells underlies the heterogeneity observed in HB.” Could the authors please comment.

We thank this reviewer for these comments. As appropriately stated, after subclustering on the 6 tumor cell clusters, we identified 5 distinct tumor signatures. We agree with this reviewer that the neuroendocrine signature was almost exclusively found in patient 8 with some tumor cells of

patient 7 exhibiting moderate levels of neuroendocrine signature expression. The remaining four HB signatures were expressed to varying degrees in multiple patients (Fig. 2c-f). The variable amount of each HB signature present in each tumor supports our conclusion that the relative proportions of each tumor signature underlies the heterogeneity observed in HB. We do acknowledge that our wording and conclusion may have been too strong and have therefore softened our conclusion in the Discussion (page 15, paragraph 1).

8. The authors compare their data to bulk tissue gene expression datasets. However, as there is a scRNA-Seq dataset available it would be more relevant to see how the authors data compare to this.

We thank the reviewer for the comment. We extracted the single-cell RNA-seq object from Bondoc et al.³ (<https://doi.org/10.1038/s42003-021-02562-8>) and subset the 34,307 tumor cells from the three HB tumors included in this study. We tested our HB tumor signatures and found that two of the three HB patients in the Bondoc et al. study showed enrichment for Hepatoblast-I and Hepatoblast-II signatures. We did not identify enrichment for the other three signatures (Erythroid-like, DCN-high, and Neuroendocrine). These two patients from Bondoc et al.³ also corresponded to the Hirsch et al.² hepatic differentiation classification.

The remaining hb17-tumor sample included in the Bondoc et al. study did not show enrichment in any of the five HB signatures described in our study (see above) or the other established HB classifications derived from the Hirsch et al.² (*proliferation, mesenchymal, progenitor*) and

Nagae et al¹. (*mesenchymal, hepatocyte and proliferative*) bulk RNA-seq studies. Tumor cells in sample hb17-tumor expressed higher levels of genes such as CST4, STPG2, ENPP2 and AGLB4 (see heatmap below).

9. For the MARCO ligand-receptor pairs, transcriptome data is suggestive of interaction but not definitive proof. If the authors want to claim interaction stainings or other validation is necessary.

We agree with this reviewer's comments and attempted to validate ligand-receptor interactions using FISH. Due to technical limitations and inconsistent staining, we were unable to demonstrate these interactions with stains. As a result we have moved the receptor-ligand interaction in supplementary data (Supplementary Fig. 14). Our stains did, however, confirm the absence of MARCO expression on tumor macrophages (Fig. 3). Further analysis using pseudotime supports the origin of these tumor associated macrophages to be from circulating monocytes as opposed to liver-resident macrophages (i.e. Kupffer cells). This analysis has been included in a revised version of Fig. 3 and the Results (page 9, line 26).

10. Validation of the five tumour cell types by immunohistochemistry/immunofluorescence or in situ RNA hybridisation would greatly improve the quality of the manuscript.

We thank the reviewer for this comment and have revised Fig. 2 to include FISH and IF staining for markers of the five tumor cell types we have identified. In this panel, we show high levels of *IGF2* in patients with hepatoblastoma I and II signatures (patients 4 and 6), *GATA1* in patients with an erythroid like signature (patient 2), *POSTN* in DCN-high tumor signatures (patient 3), and ChgA staining in neuroendocrine tumor signature (patient 8).

11. How do the spheroids compare to healthy tissue? In some cases, healthy tissue can take over to 'contaminate' the culture if the tumours are slow growing. Can be one of the reasons for the relatively low correlation between spheroid and tumour tissue in Fig. 5d.

We thank this reviewer for raising this question. Normal hepatocytes do not grow in our culture conditions. We do however see growth of biliary epithelial cells and fibroblasts, however these cultures persist for no more than 3 passages, at which point they are outgrown by PDS. An example of these co-existing cell types (BECs, fibroblasts, and PDS) is shown in Supplementary Fig. 18c.

12. More data regarding the spheroids would improve the manuscript.

We appreciate this comment and have included additional data about our PDS (Supplementary Figs. 18-21). We have also included a table of differentially expressed genes from our PDS (Supplementary Data 6). These data demonstrate that PDS from patient 6 expresses higher levels of the Hepatoblast-like markers, *AFP* and *ALB*, and tumor markers, such as *DLK1* and *MEG3*. PDS from Patient 8 expresses the neuroendocrine marker *CHGB* and *CHGA*. PDS from Patient 7 have a higher expression of the tumor markers *GPC3* and *APCDD1*, while PDS from patient 2 express cholangioblastic markers such as *EPCAM*, *CEACAM5* and *CEACAM6*.

13. Chromosomal status

We appreciate this reviewer's comment. In order to provide additional information regarding chromosomal status of the HB PDS, we would need to perform WGS which we have not performed due to time and cost. However, we hope that the additional PDS analysis we have provided in our revised manuscript (H&E, brightfield microscopy, DEGs, growth kinetics, Wnt target genes, and *CTNNB1* mutations) will satisfy this reviewer's request for additional PDS data.

14. How often they can be passaged?

Tumor spheroids can be passaged for more than 15 times and about 6 months. These data have been included in the methods section of the manuscript.

15. *Brightfield images of late and early passages to show the consistent morphology and growth of the cultures*

Brightfield images of early and late passages have been included in Fig. 5 (early passage) and Supplementary Fig. 18 (later passage). These data show that patient specific PDS morphology is maintained over passages for every patient. Briefly, PDS from patient 2 are clustered tightly with large lumens; PDS from patient 7 and 8 are also tightly clustered, without visible lumens; and PDS from patient 6 show less dense clustering, with individual cells that are visible.

16. *Histological analysis such as H&E of the spheroids to show that they recapitulate the primary tumour tissue*

We thank the reviewer for this comment and have included these data as Supplementary Figure 18d.

17. *The spheroids have CTNNB1 mutations. Do the authors see an increased expression of Wnt target genes?*

We appreciate this comment and have generated violin plots showing that the expression of four common Wnt targets (AXIN2, NKD1, APCDD1 and NOTUM) was higher in tumor PDS lines in comparison to cultivated cholangiocytes and fibroblasts.

18. *The authors should show sanger sequencing traces of both the primary tumour tissues and spheroids confirming the CTNNB1 mutations*

We appreciate this reviewer's comment and have analyzed sanger sequencing traces of the majority of primary tumor tissues (patients 2, 3, 4, 6, 7, and 8) and spheroids (when available). Heterozygous CTNNB1 point mutations were found for patients 3 and 6, while patients 2, 6, 7 and 8 had deletions. Mutation details are shown in a new supplementary figure (Supplementary Fig. 8a).

19. *Did the authors establish hepatocyte organoids from the healthy tissue? Being able to perform drug screens on tumour spheroids and matched healthy organoids would be a valuable addition to the manuscript.*

We agree that growing matched hepatocyte organoids from healthy tissue would be valuable. We have attempted to establish organoids from adjacent normal tissue using the same culture conditions used to grow tumor spheroids and have been unsuccessful. Further work will be needed to identify culture conditions that are ideal for both tumor and healthy tissue spheroid cultures.

Minor points

1. *Add the following info to the table summarising the patient information: whether there is matched normal tissue or not, the total number of cells and the number of cells passing quality control (can still keep the bar graph).*

We revised Supplementary Data 2 to include the matched normal/tumor information and the number of cells passing QC cutoffs.

2. *“One of the nine patients had tumor extension into the adjacent tissue and therefore non-tumor tissue was unavailable.” – state which patient in the main text.*

Patient 5 had tumor extension into adjacent tissue and therefore non-tumor tissue from this patient was not available. This detail has been included in the main text of the manuscript. (page 3, line 26).

3. *For all the violin plots do box plots within the violin plots rather than single dots, as it is not possible to see the shape of the violin plots due to the high number of dots.*

We apologize for the unclear violin plots. In the revised manuscript, for violin plots with a high number of dots, we generated box plots within the violin plots so that the shape of the violin body is now clearly seen.

4. *Fig. 1f. Does the patient contribution to each cluster reflect the proportion of cells (relative to the number of cells from the other patients) coming from that patient? Could the authors please comment.*

We thank the reviewer for this question and have addressed this question by examining the total non-tumor or non-tumor-associated populations for each patient. The contribution of certain cell populations correlated with the total number of cells from a particular patient. For example, the decreasing proportion of NK cells in patients 4, 8, and 6 mirrored the relative decrease in total cells in these patients (patient 4>8>6). Similar findings were noted in neutrophils among patients 1, 6, and 8 where the proportion of neutrophils reflected the total cells from these patients. However, this did not hold true for all cell types and for all patients. For example, there was an increase in T cells in patient 8 compared to patient 4 despite fewer total cells obtained from patient 8. Similarly, patient 2 had higher numbers of smooth muscle cells compared to patients with higher total cell numbers. We speculate that the lack of consistent correlation of cell proportions with that of total cells contributed by a particular patient is likely related to the histological differences between HB samples and variations in local tumor microenvironments.

5. Fig. 1b. do a legend with the text rather than writing it out in the UMAP. It is difficult to read.

We fixed the UMAP by listing the cell annotations in a figure legend instead of labeling them on the UMAP.

6. Fig2c Could the authors please draw dotted lines in the correlation heatmap to mark the different signatures and include a table summarising which tumour cell clusters are included in each signature?

We added another layer of annotation to the correlation heatmap showing the different HB signatures. And we revised Supplementary Data 3 to include a summary sheet of which tumor cell clusters are included in each signature.

7. Fig. 3k Figure legend correction -Not for each patient, but for each tumour signature. By including the MARCOHigh cells in the dot plot the difference can be seen more easily.

We thank this reviewer for this comment. This ligand-receptor analysis has been moved to Supplementary Fig. 14 and the corresponding figure legend has been corrected as suggested by this reviewer.

8. Show UMAP plots of some of the representative genes for each tumour signature.

We generated featureplots for the four top markers in each HB signature. These data have been included as a separate supplementary figure (Supplementary Fig. 11)

9. With regards to the authors analysis of erythroid cells, have the authors tried to add some of the factors secreted by these cells to their spheroid cultures to see if it affects their growth?

We appreciate this comment and suggestion however we have not yet completed these experiments.

10. PCA plots showing how tumour and matched normal samples cluster relative to each other.

We generated a PCA reduction figure and it demonstrates that among normal and tumor samples, most cells clustered together, which is consistent with our analysis that non-tumor clusters do not cluster in a patient-specific manner.

11. Stainings of marker genes of the spheroid cultures

We thank the reviewer for this comment and agree that stainings of PDS cultures would be an interesting addition to our manuscript. However, we have been unable to perform stains due to the cost and time required to complete these experiments. We have intentionally prioritized the tissue stains included in revised Figures 2, 3 and 4. We do agree, however, that additional information about PDS cultures is beneficial and have completed H&E stains (Supplementary Fig. 18) of PDS from patients 2, 6 and 8, and have compared PDS morphology to the tumor tissues for those patients. In addition, we have added differentially expressed genes of the four PDS lines (Supplementary Data 6) highlighting patient-specific gene profiles for each PDS.

12. Supplementary Fig. 7. Use dotted lines to indicate tumour tissue.

We thank the reviewer for this comment. We have taken the opportunity to include more comprehensive H&E sections than what was included in our original manuscript. Supplementary Fig. 10 has been updated to show low and high magnification of each tumor type, along with the identification of unique features that correspond to our transcriptional analysis of each tumor. We have also delineated normal and tumor tissue with dotted lines where applicable.

13. Supplementary Fig. 9. Please also do the sub clustering of the healthy tissue and the healthy and tumour tissue together

We extracted cells from paired normal tissues and re-clustered the 16,098 cells. Using our previously established cell type annotation, we generated a UMAP (panel a and b) and computed the cell type composition from each patient (panel c). We observed a much smaller population of tumor cells ($n = 91$) from eight patients, and that all the non-tumor or non-tumor-associated populations consisted of multiple patients, validating our analysis that only tumor cells exhibited patient-specific heterogeneity.

14. *Supplementary Fig. 14b growth curves would be more informative*

We have revised our presentation of PDS proliferation as growth curves. These data are included as a revised figure (Supplementary Fig. 18b).

REFERENCES

1. Nagae, G. *et al.* Genetic and epigenetic basis of hepatoblastoma diversity. *Nat. Commun.* **12**, 1–16 (2021).
2. Hirsch, T. Z. *et al.* Integrated genomic analysis identifies driver genes and cisplatin-resistant progenitor phenotype in pediatric liver cancer. *Cancer Discov.* **11**, 2524–2543 (2021).
3. Bondoc, A. *et al.* Identification of distinct tumor cell populations and key genetic mechanisms through single cell sequencing in hepatoblastoma. *Commun. Biol.* **4**, 1–14 (2021).
4. Galluzzi, L. *et al.* Molecular mechanisms of cisplatin resistance. *Oncogene* **31**, 1869–1883 (2012).
5. Von Schweinitz, D., Schmidt, D., Fuchs, J., Welte, K. & Pietsch, T. Extramedullar hematopoiesis and intratumoral production of cytokines in childhood hepatoblastoma. *Pediatr. Res.* **38**, 555–563 (1995).
6. Wu, Y. X., Yang, J. H. & Saito, H. Bortezomib-resistance is associated with increased levels of proteasome subunits and apoptosis-avoidance. *Oncotarget* **7**, 77622–77634 (2016).
7. Zhang, J. *et al.* TRIM28 attenuates Bortezomib sensitivity of hepatocellular carcinoma cells through enhanced proteasome expression. *Clin. Transl. Med.* **12**, 2–7 (2022).

Reviewers' Comments:

Reviewer #1:

Remarks to the Author:

the authors have improved their manuscript.

Still very descriptive paper, difficult to follow but observations are more precise.

Be careful to don't transform unique cases with very specific features into "classification". In my view this is the major point that remains difficult.

Reviewer #2:

Remarks to the Author:

Thank the authors for their efforts in addressing initial comments. The authors have highly improved the manuscript. There are a few minor issues below for the authors.

1. In Fig. 1c; Fig.2b,m; Fig.3e-g; Fig. 4c,e,g,h; Fig.6d-e, Gene labels need to indicate gene expression in italics.
2. In Fig. 6a, d-f, p-values are not indicated by asterisks or specific numerical values, and statistical analysis methods are not shown in the legend.
3. In Fig. 6f refer to the previous minor comment 7, the authors showed the expression of proteasome genes among different patients. However, the proteasome genes were not consistent in patients 2 and 7, and patients 6 and 8 were not consistent. It is not good for a single gene expression and proteasome score to be on the same x-axis. The author can display the proteasome score of different patients through boxplot or vlnplot, and perform statistical analysis to show the difference. This is sufficient to explain the different degrees of proteasome enrichment in different patients.

Reviewer #3:

Remarks to the Author:

In the revised version of the manuscript entitled "Single-cell analysis of hepatoblastoma identifies tumor signatures that predict chemotherapy susceptibility using patient-specific tumor spheroids" the authors have addressed most of my comments. I appreciate the added information about the spheroid cultures and I believe the manuscript overall has improved. However, I still have a few minor questions and remarks related to the revision.

- My interpretation of the manuscript title is that the signatures can help predict treatment susceptibility.

However, if the epithelial spheroid cultures (lacking other cell types, the authors mention that the fibroblast and biliary epithelial cells do not persist for more than 3 passages) can predict treatment response, is the inclusion of the other cell types necessary, or could the signatures also be based on the epithelial cells only?

It seems like the authors mainly look at the expression data of the PDS for the drug screens – a stronger link to the signatures would be of value if the authors wish to state that the signatures can predict the chemo response. If the authors could show how the non-epithelial cell types aid the prediction of treatment response it would be a great addition that would support the title of the manuscript.

- The authors do a high QC cutoff based on gene and transcript numbers. What about the percentage of mitochondrial reads? Common cutoff is around 5%. Could the authors please comment.

- When comparing the high and low QC cutoffs it would be interesting to know if any clusters are lost. If so, how do those clusters contribute to the signatures. If low QC clusters would highly contribute to

some of the signatures it would be a sign for caution.

- Comment 11: Using the same antibodies as in fig2 might be possible – but not required.
- Comment 12: I believe the dotted lines are missing in supp fig 10. Red and yellow arrows are missing in some images. Zoomed-in inserts would be beneficial. Scalebars are missing.
- Comment 13: large chromosomal abnormalities can be detected using karyotyping.
- Throughout the manuscript the location of the scalebars need to be changed as it it sometimes difficult to see. The size indication should also be removed as it is not readable.
- For all the IF images zoomed-in inserts should be added
- Gene names are missing in the Neuroendocrine_Full_Signature excel tab
- Fig6 asterix special characters are missingthank

Point-by-point response to reviewers' comments

REVIEWER COMMENTS

Reviewer #1 (Remarks to the Author): *the authors have improved their manuscript.*

Still very descriptive paper, difficult to follow but observations are more precise.

Be careful to don't transform unique cases with very specific features into "classification". In my view this is the major point that remains difficult.

We appreciate that this reviewer finds our manuscript improved and wholeheartedly agree that findings made from unique cases does not translate to new classifications. Our identification of five tumor signatures was derived from the amalgamation of all the cells from all nine tumors in our study. Our conclusion that these five signatures are shared across patients has been supported by the main observation that these signatures are either found in more than one patient in our study or these signatures correspond to groupings established by previously published transcriptomic datasets (**Supplementary Fig. 12**). We believe that our data indicate that the signatures we have identified are features shared across patients, and may account for the heterogeneity observed in HB.

We also agree that the message of the paper can be clarified and toned down so that our findings are not overstated. We have specifically changed key language throughout the manuscript with the main changes summarized below:

1. Instead of stating that the five signatures we have identified drive tumor heterogeneity, we have used softer language and have stated that the five hepatoblastoma signatures may account for the tumor heterogeneity observed (**Abstract, page 2, line 7; Introduction, page 3, line 14; Results, page 6, line 24, and page 9, line 3; Discussion, page 15, line 17**).
2. We have eliminated the use of the word classification in the manuscript and figures in order to prevent the reader from extrapolating our findings and previously published transcriptomic data to mean that distinct HB classifications have been defined.
3. We have added an additional sentence in the discussion (**Page 15, line 20**) that reiterates that, "though only five signatures were identified in the current study, our data suggests that additional signatures need to be defined in order to comprehensively account for the heterogeneity observed in HB tumors."

We hope these revisions address this reviewer's concerns.

Reviewer #2 (Remarks to the Author):

Thank the authors for their efforts in addressing initial comments. The authors have highly improved the manuscript. There are a few minor issues below for the authors.

We appreciate for this Reviewer's favorable impression of our revised manuscript. See our point-by-point responses to address the minor issues raised below.

1. *In Fig. 1c.; Fig.2b,m; Fig.3e-g; Fig. 4c,e,g,h; Fig.6d-e, Gene labels need to indicate gene expression in italics.*

We apologize for this oversight. The figures listed above along with supplementary figures have been edited so that gene labels are now appropriately italicized.

2. *In Fig. 6a, d-f, p-values are not indicated by asterisks or specific numerical values, and statistical analysis methods are not shown in the legend.*

Thank you for pointing out our inadvertent omission of these details. The p-values in **Fig. 6a,d-f** are now indicated by asterisks and corresponding statistical methods are now included in the legend.

3. *In Fig. 6f refer to the previous minor comment 7, the authors showed the expression of proteasome genes among different patients. However, the proteasome genes were not consistent in patients 2 and 7, and patients 6 and 8 were not consistent. It is not good for a single gene expression and proteasome score to be on the same x-axis. The author can display the proteasome score of different patients through boxplot or violinplot, and perform statistical analysis to show the difference. This is sufficient to explain the different degrees of proteasome enrichment in different patients.*

We appreciate this Reviewer's thoughtful suggestion and have made the changes as suggested to display the proteasome score of different patients using a violin plot. These changes are now included in an updated **Fig. 6f**.

Reviewer #3 (Remarks to the Author):

In the revised version of the manuscript entitled "Single-cell analysis of hepatoblastoma identifies tumor signatures that predict chemotherapy susceptibility using patient-specific tumor spheroids" the authors have addressed most of my comments. I appreciate the added information about the spheroid cultures and I believe the manuscript overall has improved. However, I still have a few minor questions and remarks related to the revision.

We thank this reviewer for their review of our revised manuscript and we are appreciative that they feel that the manuscript overall has improved. We have addressed their minor questions and remarks below.

1. *My interpretation of the manuscript title is that the signatures can help predict treatment susceptibility.*

However, if the epithelial spheroid cultures (lacking other cell types, the authors mention that the fibroblast and biliary epithelial cells do not persist for more than 3 passages) can predict treatment response, is the inclusion of the other cell types necessary, or could the signatures also be based on the epithelial cells only? It seems like the authors mainly look at the expression data of the PDS for the drug screens – a stronger link to the signatures would be of value if the authors wish to state that the signatures can predict the chemo response. If the authors could show how the non-epithelial cell types aid the prediction of treatment response it would be a great addition that would support the title of the manuscript.

We appreciate this reviewer's comments and questions. To clarify, our culture conditions were optimized for the growth of tumor cells. Non-tumor epithelial cells including hepatocytes do not grow, and BECs only proliferate for a few passages. Furthermore, only specific subpopulations of tumor cells within each tumor are maintained as spheroids. Both epithelial and mesenchymal tumor cells can be grown as spheroids, as evidenced by the neuroendocrine spheroid from patient 8. However, it appears that only one tumor cell subpopulation is maintained within each spheroid. During early spheroid passages (passages 1-3), we could not generate enough spheroid cells to perform scRNAseq, thus cannot comment on whether multiple tumor subpopulations are present initially in spheroids.

In **Fig. 5** and the updated **Supplementary Fig. 18**, we show that each spheroid is similar to the specific tumor subpopulation from which each spheroid grew (termed "parent cells" in

manuscript), as well as one of the five tumor signatures (except for spheroids from patient 2) based on gene expression and staining characteristics. In **Fig. 6** and **Supplementary Fig. 21** we show that the gene expression of each spheroid, its parent cell, and the tumor signature it corresponds to can predict treatment response for that specific tumor subpopulation. Thus, our emphasis in the manuscript that PDS can help predict treatment response according to tumor signature.

2. *The authors do a high QC cutoff based on gene and transcript numbers. What about the percentage of mitochondrial reads? Common cutoff is around 5%. Could the authors please comment.*

To ensure that we capture as many usable cells for analysis as possible, we implemented a relatively low QC cutoff of 300 genes, 500 transcripts and 25% mitochondrial reads. Since human hepatocytes are known to have a much higher percentage of mitochondrial genes^{1,2}, several tumor clusters and epithelial populations showed higher percentage of MT% compared to stromal and immune cells (shown in the violin plot below).

However, if we impose a stringent MT% cutoff to our data, we would lose a majority of the tumor cells and non-tumor epithelial cells (barchart below), decreasing the total number of cells by 2/3 from 29,968 to 10,106, which would prevent us from efficiently profiling the transcriptomes of HB tumors. Notably, even when using a relatively high MT% cutoff of 25%, the differentially expressed genes we used to annotate the cell populations and characterize HB tumors did not include any MT genes, indicating that cells with high MT reads are unlikely to bias our analysis results.

3. *When comparing the high and low QC cutoffs it would be interesting to know if any clusters are lost. If so, how do those clusters contribute to the signatures. If low QC clusters would highly contribute to some of the signatures it would be a sign for caution.*

We thank this reviewer for this comment. We agree that if clusters were lost as a result of a high QC cutoff, then we should be more cautious in defining the tumor signatures in our manuscript. However, after our analysis using both high and low QC cutoffs, no clusters were lost when changing our cutoff thresholds. This supports the use of our low QC cutoff in our analysis.

4. *Comment 11: Using the same antibodies as in fig2 might be possible – but not required.*

We thank the reviewer for this suggestion and regret that we did not do this at the first revision. We have stained our spheroid cultures using FISH and IF and demonstrate that the gene expression of HB tumors and specific tumor signatures are retained in spheroids. These data have been included as an additional panel in **Supplementary Fig. 18d** and have been included in the Results (**Page 13, Line 4**).

5. *Comment 12: I believe the dotted lines are missing in supp fig 10. Red and yellow arrows are missing in some images. Zoomed-in inserts would be beneficial. Scalebars are missing.*

We thank this reviewer for this comment and apologize for not including the dotted lines in the revised **Supplementary Fig. 10**. These lines have now been included where applicable as some images are only displaying tumor without adjoining normal liver. We have also included additional red and yellow arrows where applicable as not all specimens have specific features that need to be highlighted. Finally, we have intentionally provided both full-size low-power and high-power images for all patients as opposed to zoomed-in inserts. Any further magnification than we have already provided (i.e. greater than 400X) would be a digital magnification as opposed to a true optical magnification. We do appreciate this reviewer's comment that clearer indication of the portions of low-power images that are displayed at high power would be of benefit, and have addressed this concern by denoting areas of the low-power tissue section that are being shown at a higher power. Finally, we have added scalebars as requested by this reviewer.

6. *Comment 13: large chromosomal abnormalities can be detected using karyotyping.*

We thank this Reviewer for this comment and agree that our understanding of our spheroid cultures would be enhanced if we can detect large chromosomal abnormalities via a karyotype analysis. Though karyotyping analysis is well established for 2D cultures, we have inquired about the feasibility of this analysis and find that completing this analysis of our 3D HB organoids will require both experimental optimization and a longitudinal analysis of our cultures³. We believe the analysis required to establish chromosomal stability for our spheroids is outside the scope of the current study. However, we appreciate that this Reviewer has brought this point to our attention, and plan to conduct this analysis in the future.

7. *Throughout the manuscript the location of the scalebars need to be changed as it is sometimes difficult to see. The size indication should also be removed as it is not readable.*

We thank this reviewer for this comment and have added scalebars throughout the manuscript. The size indications have also been removed.

8. *For all the IF images zoomed-in inserts should be added*

Zoomed-in inserts have been added for the following figures: **Figs. 3 and 4, and Supplementary Figs. 6 and 10.**

9. *Gene names are missing in the Neuroendocrine_Full_Signature excel tab*

We apologize for any confusion this caused. The gene names were listed in Column F instead of Column A. This has been fixed in an updated **Supplementary Data 4.**

10. *Fig6 asterix special characters are missingthank*

Thank you for this comment. Asterisks indicating significance have now been added to **Fig. 6.**

References

1. Osorio D, Cai JJ. Systematic determination of the mitochondrial proportion in human and mice tissues for single-cell RNA-sequencing data quality control. *Bioinformatics* **37**, 963967 (2021).
2. MacParland SA, *et al.* Single cell RNA sequencing of human liver reveals distinct intrahepatic macrophage populations. *Nat Commun* **9**, 4383 (2018).
3. Bolhaqueiro ACF, *et al.* Ongoing chromosomal instability and karyotype evolution in human colorectal cancer organoids. *Nat Genet* **51**, 824-834 (2019).

Reviewers' Comments:

Reviewer #3:

Remarks to the Author:

In the second round of revision the authors have again improved their manuscript. I thank the authors for their explanation regarding their culture conditions.

I only have two final remarks, where I find the second remark most important.

In supp fig18e a few cells from the spheroids derived from patient 8 cluster with the endo/fibro_like_tumour clusters. I doubt that these few non-epithelial like cells has a strong influence on the drug response. Moreover, the presence of the mesenchymal cells in the spheroid culture is not evident. Could the authors please provide stainings if they want to further support this claim. Please also indicate passage numbers in the figure legend – as early and late is just relative.

Drug resistance is quantified as IC50 and Emax, or alternatively GR50 and GRmax. I believe that analysing and displaying the drug screen data in Fig 6 according to the GR metrics (<http://www.grcalculator.org/grcalculator/>) would make it clearer and more easy to digest for the reader. It would also align with what are the common standards in the field.

I apologise for not mentioning this in the previous round of revision. However, since it only requires 're-analysis' of already acquired data I do believe it is something the authors can easily accomplish. It would significantly improve the manuscript.

Point-by-point response to reviewers' comments

REVIEWERS' COMMENTS

Reviewer #3 (Remarks to the Author):

In the second round of revision the authors have again improved their manuscript. I thank the authors for their explanation regarding their culture conditions.

I only have two final remarks, where I find the second remark most important.

We thank this Reviewer for their thoughtful comments and appreciate that they find that our manuscript has improved. We have addressed this Reviewer's specific remarks below.

1. In supp fig18e a few cells from the spheroids derived from patient 8 cluster with the endo/fibro_like_tumour clusters. I doubt that these few non-epithelial like cells has a strong influence on the drug response. Moreover, the presence of the mesenchymal cells in the spheroid culture is not evident. Could the authors please provide stainings if they want to further support this claim. Please also indicate passage numbers in the figure legend – as early and late is just relative.

We thank the reviewer for this comment. We believe that this Reviewer is referring to supp figure 19e, and not 18e. We do not appreciate any endo/fibro_like_tumor cells that cluster close to spheroids for patient 8. We do, however, observe a cluster of Neuroendocrine tumor cells that are adjacent to the spheroid cluster (one cluster of tumor cells have a signature score of 0.2 for the Neuroendocrine signature).

e. Patient 8

With regard to this Reviewer's comment regarding mesenchymal cells, fibroblasts were only identified in spheroids derived from patients 7 and 9 (supp figure 19d, "Spheroids_2" for patient 7, and supp figure 19f, "Spheroids_1" for patient 9) and by brightfield images (Supp figure 18c). We also agree that it is extremely unlikely that fibroblasts found in spheroid cultures impact drug response, especially because the number of fibroblasts were outnumbered by tumor cells, and fibroblasts disappeared from the culture after ~3 passages. Finally, details regarding passage numbers are now included in the corresponding figure legends (Figure 5, Supp. Figures 18 and 19).

2. Drug resistance is quantified as IC50 and Emax, or alternatively GR50 and GRmax. I believe that analysing and displaying the drug screen data in Fig 6 according to the GR metrics (<http://www.grcalculator.org/grcalculator/>) would make it clearer and more easy to digest for the reader. It would also align with what are the common standards in the field. I apologise for not mentioning this in the previous round of revision. However, since it only requires 're-analysis' of already acquired data I do believe it is something the authors can easily accomplish. It would significantly improve the manuscript.

We thank the reviewer for this comment, and agree that showing IC50 results would be clearer and would follow common standards in the field. We calculated the IC50 for each drug tested and replaced the panels in Figure 6 that show drug responses by patient-specific spheroids. We have also generated a new supplementary figure (Supp. Fig. 21) that includes the drug response graphs that were originally included in Figure 6.